# QT-DoG: Quantization-aware Training for Domain Generalization

## Abstract

Domain Generalization (DG) aims to train models that perform well not only on the training (source) domains but also on novel, unseen target data distributions. A key challenge in DG is preventing overfitting to source domains, which can be mitigated by finding flatter minima in the loss landscape. In this work, we propose Quantization-aware Training for Domain Generalization (QT-DoG) and demonstrate that weight quantization effectively leads to flatter minima in the loss landscape, thereby enhancing domain generalization. Unlike traditional quantization methods focused on model compression, QT-DoG exploits quantization as an implicit regularizer by inducing noise in model weights, guiding the optimization process toward flatter minima that are less sensitive to perturbations and overfitting. We provide both an analytical perspective and empirical evidence demonstrating that quantization inherently encourages flatter minima, leading to better generalization across domains. Moreover, with the benefit of reducing the model size through quantization, we demonstrate that an ensemble of multiple quantized models further yields superior accuracy than the state-of-the-art DG approaches with no computational or memory overheads. Our extensive experiments demonstrate that QT-DoG generalizes across various datasets, architectures, and quantization algorithms, and can be combined with other DG methods, establishing its versatility and robustness.

## 1 Introduction

Many works have shown that deep neural networks trained under the assumption that the training and test samples are drawn from the same distribution fail to generalize in the presence of large training-testing discrepancies, such as texture (Geirhos et al., 2019; Bahng et al., 2020), background (Xiao et al., 2020), or day-to-night (Dai & Van Gool, 2018; Michaelis et al., 2019) shifts. Domain Generalization (DG) addresses this problem and aims to learn models that perform well not only in the training (source) domains but also in new, unseen (target) data distributions (Blanchard et al., 2011; Muandet et al., 2013; Zhou et al., 2022).

In the broader context of generalization, with training and test data drawn from the same distribution, the literature has revealed a relationship between the flatness of the loss landscape and the generalization ability of deep learning models (Keskar et al., 2017; Dziugaite & Roy, 2017; Garipov et al., 2018; Izmailov et al., 2018; Jiang et al., 2020; Foret et al., 2021; Zhang et al., 2023). This relationship has then been leveraged by many recent works, demonstrating that a flatter minimum also improves Out-of-Distribution (OOD) performance (Cha et al., 2021; Ramé et al., 2023; Arpit et al., 2022). At the heart of all these DG methods lies the idea of weight averaging (Izmailov et al., 2018), which involves averaging weights from several trained models or at various stages of the training process.

In this work, we demonstrate that flatter minima in the loss landscape can be effectively achieved through weight quantization using Quantization-aware Training (QAT), making it an effective approach for DG. By restricting the possible weight values to a lower bit precision, quantization imposes constraints on the weight space, introducing quantization noise into the network parameters. This noise, as discussed in prior works (An, 1996; Murray & Edwards, 1992; Goodfellow et al., 2016; Hochreiter & Schmidhuber, 1994), acts as a form of regularization that naturally encourages the optimization process to converge toward flatter minima. Furthermore, our results show that models trained with quantization not only generalize better across domains but also reduce overfitting to source domains. To the best of our knowledge, this is the first work to explicitly explore the intersection of

Figure 1: **Performance Comparison on the Domainbed Benchmark.** We show the average accuracy on 5 different datasets. *One Model* refers to methods training a single ResNet-50 model. *Multiple Models* refers to training *M* models for averaging or ensembling, which affects the training cost. We compare QT-DoG and EoQ to other state-of-the-art methods. The marker size is proportional to the memory footprint. EoQ shows superior performance despite being 4 times smaller than its full-precision counterpart. Additionally, QT-DoG demonstrates comparable performance to *One Model* methods, despite its significantly smaller size.

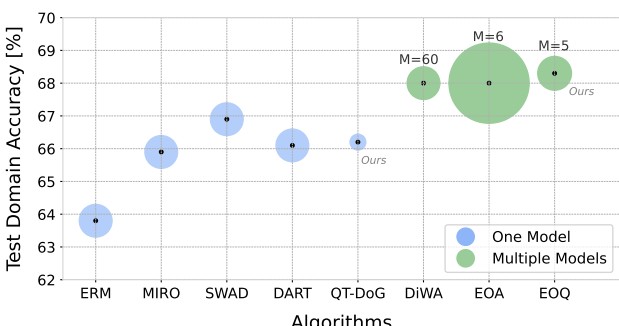

quantization and domain generalization. Through both analytical reasoning and empirical validation, we provide strong evidence that QAT promotes flatter minima, leading to enhanced generalization performance on unseen domains.

The benefit of having fast and light-weight quantized models then further allow us to even make an ensemble of them, termed *Ensemble of Quantization* (EoQ). EoQ achieves superior performance while maintaining the computational efficiency of a single full-precision model. This stands in contrast to ensemble-based methods like (Ramé et al., 2023; Arpit et al., 2022), which require storing and running multiple full-precision models. With our approach, quantization not only improves generalization but also reduces the model's memory footprint and computational cost at inference. As shown in Figure 1, EoQ yields a model with a memory footprint similar to the state-of-the-art single-model DG approaches and much smaller than other ensemble-based methods, yet outperforms all its competitors in terms of accuracy.

Our contributions can be summarized as follows:

- **Quantization for Domain Generalization.** We are the first to demonstrate that quantization-aware training, traditionally used for model compression, can serve as an implicit regularizer, with quantization noise enhancing domain generalization.

- **Flat Minima via QAT:** We empirically demonstrate that QAT promotes flatter minima in the loss landscape and provide an analytical perspective behind this effect. Additionally, we show that QAT stabilizes model behavior on OOD data during training.

- **Resource Efficiency:** In contrast to traditional DG methods that often increase model size or computational cost, QT-DoG not only improves generalization but also significantly reduces the model size, enabling efficient deployment in real-world applications. EoQ, for instance, requires nearly 6 times less memory than Arpit et al. (2022) and 12 times less training compute compared to Ramé et al. (2023), which trains 60 models for diverse averaging.

- **EoQ** We introduce EoQ, a strategy that combines the power of quantization and ensembling for generalization. EoQ achieves state-of-the-art results on the DomainBed benchmark while matching the computational cost and memory footprint as a single full-precision model.

Our code and models will be made publicly available.

## 2 RELATED WORK

### 2.1 DOMAIN GENERALIZATION

Numerous multi-source domain generalization (DG) methods have been proposed in the past. In this section, we review some of the recent approaches, categorizing them into different groups based on their methodologies.

### 2.1.1 Domain Alignment

The methods in this category focus on reducing the differences among the source domains and learn domain-invariant features (Arjovsky et al., 2019; Krueger et al., 2021; Rame et al., 2022a; Sun et al., 2016; Sagawa et al., 2020; Ganin et al., 2016; Li et al., 2023; Cheng et al., 2024). The core idea is that, if the learnt features are invariant across the different source domains, they will also be robust to the unseen target domain. For matching feature distributions across source domains, DANN (Ganin et al., 2016) uses an adversarial loss while CORAL (Sun & Saenko, 2016) and DICA (Muandet et al., 2013) seek to align latent statistics of different domains. Unfortunately, most of these methods fail to generalize well and were shown not to outperform ERM on various benchmarks (Gulrajani & Lopez-Paz, 2021; Ye et al., 2022; Koh et al., 2021a).

### 2.1.2 Regularization

In the literature, various ways of regularizing models (implicit and explicit) have also been proposed to achieve better generalization. For example, invariant risk minimization (Arjovsky et al., 2019) relies on a regularization technique such that the learned classifier is optimal even under a distribution shift. Moreover, (Huang et al., 2020) tries to suppress the dominant features learned from the source domain and pushes the network to use other features correlating with the labels. Furthermore, (Krueger et al., 2021) proposes risk extrapolation that uses regularization to minimize the variance between domain-wise losses, considering that it is representative of the variance including the target domain.

### 2.1.3 Vision Transformers

Recent studies have increasingly utilized vision transformers for domain generalization (Shu et al., 2023; Sultana et al., 2022). Some approaches enhance vision transformers by integrating knowledge distillation (Hinton et al., 2015) and leveraging text modality from CLIP (Radford et al., 2021) to learn more domain-invariant features (Moayeri et al., 2023; Addepalli et al., 2024; Chen et al., 2024; Huang et al., 2023; Liu et al., 2024).

### 2.1.4 Ensembling

Ensembling of deep networks (Lakshminarayanan et al., 2017; Hansen & Salamon, 1990; Krogh & Vedelsby, 1995) is a foundational strategy and has consistently proven to be robust in the past. Many works have been proposed to train multiple diverse models and combine them to obtain better in-domain accuracy and robustness to domain shifts (Arpit et al., 2022; Thopalli et al., 2021; Mesbah et al., 2022; Li et al., 2022; Lee et al., 2022; Pagliardini et al., 2023). However, ensembles require multiple models to be stored and a separate forward pass for each model, which increases the computational cost and memory footprint, especially if the models are large.

### 2.1.5 Weight Averaging

Combining or averaging weights from different training stages or models has emerged as a robust approach to improve OOD generalization (Wortsman et al., 2022b; Matena & Raffel, 2022; Wortsman et al., 2022a; Gupta et al., 2020; Choshen et al., 2022; Wortsman et al., 2021; Maddox et al., 2019; Benton et al., 2021; Cha et al., 2021; Jain et al., 2023; Ramé et al., 2023). Techniques like SWAD (Cha et al., 2021) leverage weight averaging to identify flat minima, reducing overfitting and enhancing generalization under distribution shifts. Similarly, DiWA (Rame et al., 2022b) combines weights from independently trained models to improve robustness through increased diversity.

Arpit et al. (2022) integrates ensembling with weight averaging, yielding superior performance compared to either method alone, albeit with significant memory and computational costs. To address these challenges, we demonstrate that quantization can improve generalization while reducing resource demands.

Although flatter minima are not universally indicative of better domain generalization (Andriushchenko et al., 2023), they remain a valuable tool for improving robustness in many scenarios. Moreover, recent findings (Mueller et al., 2023) highlight that selective application of SAM (Foret et al., 2021), such as restricting it to normalization layers, can further refine its effectiveness. The consistent empirical success of SAM underscores its reliability as a method for enhancing domain generalization, despite the nuanced relationship between flatness and performance across different settings.

## 2.2 MODEL QUANTIZATION

Model quantization is used in deep learning to reduce the memory footprint and computational requirements of deep network. In a conventional neural network, the model parameters and activations are usually stored as high-precision floating-point numbers, typically 32-bit or 64-bit. The process of model quantization entails transforming these parameters into lower bit-width representations, such as 8-bit integers or binary values. Existing techniques fall into two main categories. **Post-Training Quantization** (PTQ) quantizes a pre-trained network using a small calibration dataset and is thus relatively simple to implement (Nagel et al., 2020; Li et al., 2021; Frantar & Alistarh, 2022; Zhao et al., 2019; Cai et al., 2020; Nagel et al., 2019; Shao et al., 2024; Lin et al., 2024; Chee et al., 2023). **Quantization-Aware Training** (QAT) retrains the network during the quantization process and thus better preserves the model's full-precision accuracy. Yang et al. (2023); Esser et al. (2020); Zhou et al. (2017); Bhalgat et al. (2020); Yamamoto (2021); Yao et al. (2020); Shin et al. (2023). In the next section, we provide some background on quantization and on the method we will use in our approach. Our goal in this work is not to introduce a new quantization strategy but rather to demonstrate the impact of quantization on generalization.

## 3 DOMAIN GENERALIZATION BY QUANTIZATION

We build our method on the simple ERM approach to showcase the effects of quantization on the training process and on the generalization to unseen data from a different domain. Despite the simplicity of this approach, we will show in Section 4.3 that it yields a significant accuracy boost on the test data from the unseen target domain. Furthermore, it stabilizes the behavior of the model on OOD data during training, making it similar to that on the in-domain data. In the remainder of this section, we focus on providing some insights on how quantization enhances DG.

### 3.1 QUANTIZATION

Let $w$ be a single model weight to be quantized, $s$ the quantizer step size, and $Q_N$ and $Q_P$ the number of negative and positive quantization levels, respectively. We define the quantization process that computes $\bar{w}$, a quantized and integer scaled representation of the weights, as

$$\bar{w} = \lfloor clip(w/s, -Q_N, Q_P) \rceil, \tag{1}$$

where the function $clip(k, r_1, r_2)$ is defined as

$$clip(k, r_1, r_2) = \begin{cases} \lfloor k \rceil & \text{if } r_1 < k < r_2 \\ r_1 & \text{if } k \leq r_1 \\ r_2 & \text{if } k \geq r_2 \end{cases} \tag{2}$$

Here, $\lfloor k \rceil$ represents rounding $k$ to nearest integer. If we quantize a weight to $b$ bits, for unsigned data $Q_N = 0$ and $Q_P = 2^b - 1$, and for signed data $Q_N = 2^{b-1}$ and $Q_P = 2^{b-1} - 1$.

Note that the quantization process described in Eq. 1 yields a scaled value. A quantized representation of the data at the same scale as $w$ can then be obtained as

$$w_q = \bar{w} \times s. \tag{3}$$

This transformation results in a discretized weight space that inherently introduces noise. We demonstrate generalization ability of QT-DoG with different quantization methods in section 4.3.4.

### 3.2 QUANTIZATION LEADS TO FLAT MINIMA

In the literature (Rame et al., 2022b; Arpit et al., 2022; Krueger et al., 2021; Cha et al., 2021; Rame et al., 2022b; Foret et al., 2021), it has been established that a model's generalization ability can be increased by finding a flatter minimum during training. This is the principle we exploit in our work, but from the perspective of quantization, and provide an analytical view into how it contributes to achieving flatter minima. In practice, ERM can have several solutions with similar training loss values but different generalization ability. Even when the training and test data are drawn from the same distribution, the standard optimizers, such as SGD and Adam (Kingma & Ba, 2015), often lead to sub-optimal generalization by finding sharp and narrow minima (Keskar et al., 2017; Dziugaite & Roy, 2017; Garipov et al., 2018; Izmailov et al., 2018; Jiang et al., 2020; Foret et al., 2021). This has been shown to be prevented by introducing noise in the model weights during training (An, 1996;

Murray & Edwards, 1992; Goodfellow et al., 2016; Hochreiter & Schmidhuber, 1994). Here, we argue that quantization inherently induces such noise and thus helps to find flatter minima.

Let $\hat{y}_i = f(\boldsymbol{x}, \boldsymbol{w})$ represent the predicted output of the network $f$, which is parameterized by the weights $\boldsymbol{w}$. A quantized network can then be represented as

$$f(\boldsymbol{x}, \boldsymbol{w}_q) = f(\boldsymbol{x}, \boldsymbol{w} + \Delta) = \hat{y}_i{}^q,$$

where $\boldsymbol{w}_q$ denotes the quantized weights and $\hat{y}_i{}^q$ the corresponding prediction. The quantized weights can thus be thought of as introducing perturbations ($\Delta$) to the full-precision weights, akin to noise affecting the weights.

Such noise induced by the weight quantization can also be seen as a form of regularization, akin to more traditional methods. For small perturbations, (An, 1996; Murray & Edwards, 1992; Goodfellow et al., 2016) show that this type of regularization encourages the parameters to navigate towards regions of the parameter space where small perturbations of the weights have minimal impact on the output, i.e., flatter minima.

When noise is introduced via quantization, second-order Taylor series approximation of the loss function for the perturbed weights $\boldsymbol{w} + \Delta$ can be expressed as

$$\mathcal{L}(\boldsymbol{w} + \Delta) \approx \mathcal{L}(\boldsymbol{w}) + \nabla\mathcal{L}(\boldsymbol{w})^\top \Delta + \frac{1}{2}\Delta^\top \mathcal{H}\Delta, \tag{4}$$

where $\mathcal{L}(\boldsymbol{w})$ is the loss at the original weights $\boldsymbol{w}$, $\nabla L(\boldsymbol{w})$ is the gradient of the loss at $\boldsymbol{w}$, and $\mathcal{H} = \nabla^2 L(\boldsymbol{w})$ is the Hessian matrix, which contains second-order partial derivatives of the loss function with respect to the weights, representing the curvature of the loss surface.

Eq. 4 shows how the quantization noise $\Delta$ interacts with the curvature $\mathcal{H}$ of the loss function. In regions with large curvature (sharp minima), the Hessian $\mathcal{H}$ has large eigenvalues, and even small perturbations $\Delta$ result in large increases in the loss (Dinh et al., 2017). In contrast, in flat regions (small eigenvalues of $\mathcal{H}$), the loss remains nearly unchanged for small perturbations. Quantization noise acts as an implicit regularizer by introducing perturbations $\Delta$ that disrupt the model's weight updates. In sharper minima, where the Hessian $\mathcal{H}$ eigenvalues are large, small noise significantly increases the loss, causing the model to "escape" these regions and search for flatter, more stable minima. In flatter regions, where the Hessian $\mathcal{H}$ eigenvalues are small, the noise has less impact, helping the model settle into these regions with lower loss. This encourages convergence to solutions that are less sensitive to small changes in the input or model parameters, which is beneficial for out-of-distribution (OOD) generalization.

In the case of quantization-aware training, the induced noise $\Delta$ is influenced by the quantization bin width or the quantizer step size $s$, and thus ranges between $-\frac{s}{2}$ and $+\frac{s}{2}$. This $s$ is directly dependent on the quantization levels or the bit-width chosen for weight quantization. As the number of bits per weight decreases, the amount of induced noise increases. Hence, the impact of the additional noise can be weighed by choosing an optimal bit-width. As will be shown in Section 4, certain bit-widths thus yield better and flatter minima that enhance generalization. However, if we induce too much noise(very low bit-precision), it introduces over-regularization. This excessive noise can overly restrict the search space, preventing the model from reaching a good solution. Instead, the optimization process may focus on minimizing the loss in a way that avoids sharp regions, but sacrifices the ability to find true minimum of the loss function. This is also evident in Table 10 in the appendix.

Moreover, Rissanen (1978); Hochreiter & Schmidhuber (1997) show that a flatter minimum corresponds to a low complexity network and requires fewer bits of information per weight. More importantly, Hochreiter & Schmidhuber (1997) demonstrates the importance of the bit-precision of the network weights and adds a regularization term in the loss function that seeks to lower the weight bit-precision to lead to flatter minima. Here, by using quantization, we are explicitly reducing bit-precision of the network weights, thus achieving the same goal.

### 3.3 EMPIRICAL ANALYSIS OF QUANTIZATION-AWARE TRAINING AND FLATNESS

In this section, we demonstrate that a flatter minimum is reached when incorporating quantization in the ERM process. Similar to (Dinh et al., 2017; Cha et al., 2021), we interpret flat minima as "a

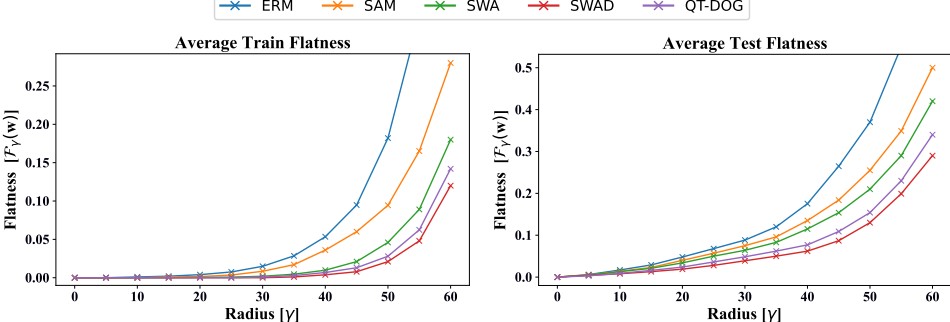

Figure 2: **Local Flatness Comparison:** We plot the average training (left) and testing (right) local flatness $\mathcal{F}\gamma(\boldsymbol{w})$ (Eq. 5) for ERM (Gulrajani & Lopez-Paz, 2021), SAM (Foret et al., 2021), SWA (Izmailov et al., 2018) and SWAD (Cha et al., 2021) by varying the radius $\gamma$ on different domains of PACS. We evaluate the training flatness $\mathcal{F}\gamma^S(\boldsymbol{w})$ on the seen domains (left) and the test flatness $\mathcal{F}_\gamma^T(\boldsymbol{w})$ on the unseen domains (right).

large connected region in weight space where the error remains approximately constant," as defined by (Hochreiter & Schmidhuber, 1997). Our loss flatness analysis shows that QT-DoG can find a flatter minimum in comparison to not only ERM but also SAM (Foret et al., 2021) and SWA (Izmailov et al., 2018).

Following the approach in Cha et al. (2021), we quantify local flatness $\mathcal{F}_\gamma(\boldsymbol{w})$ by measuring the expected change in loss values between a model with parameters $\boldsymbol{w}$ and a perturbed model with parameters $|\boldsymbol{w}'| = |\boldsymbol{w}| + \gamma$, where $\boldsymbol{w}'$ lies on a sphere of radius $\gamma$ centered at $\boldsymbol{w}$.. This is expressed as

$$\mathcal{F}_\gamma(\boldsymbol{w}) = \mathbb{E}_{\|\boldsymbol{w}'\|}[\mathcal{E}(\boldsymbol{w}') - \mathcal{E}(\boldsymbol{w})], \tag{5}$$

where $\mathcal{E}(\boldsymbol{w})$ denotes the accumulated loss over the samples of potentially multiple domains. For our analysis, we will evaluate flatness in both the source domains and the target domain, and thus $\mathcal{E}(\boldsymbol{w})$ is evaluated using either source samples or target ones accordingly.

As in Cha et al. (2021), we approximate $\mathcal{F}_\gamma(\boldsymbol{w})$ by Monte-Carlo sampling with 100 samples. In Figure 2, we compare the $\mathcal{F}_\gamma(\boldsymbol{w})$ of QT-DoG to that of ERM (Gulrajani & Lopez-Paz, 2021), SAM (Foret et al., 2021), SWA (Izmailov et al., 2018) and SWAD (Cha et al., 2021) for different radii $\gamma$. QT-DoG not only finds a flatter minimum than ERM, SAM and SWA but also yields a comparable flatness to SWAD's despite being 75% smaller in model size.

### 3.4 STABLE TRAINING PROCESS

Here, we demonstrate the robustness of out-of-domain performance to model selection using the in-domain validation set. Specifically, we seek to show that accuracy on the in-domain validation data is a good measure to pick the best model for out-of-domain distribution. Therefore, we assume that during training, the model selection criterion based on this validation data can select the best model for the OOD data even if the model starts to overfit. In other words, it is expected that the out-of-domain evaluation at each point of the training phase should improve or rather stay stable if the model is close to overfitting to the in-domain data. For these experiments, we use the TerraIncognita dataset (Beery et al., 2018) and consider the same number of iterations as for the DomainBed protocol (Gulrajani & Lopez-Paz, 2021).

As can be seen in Figure 3, vanilla ERM (without quantization) quickly overfits to the in-domain validation/training dataset. That is, the OOD performance is highly unstable during the whole training process. By contrast, our quantized model is much more stable. Specifically, we quantize our model at 2000 steps, and it can be seen that the model performance on out-of-domain distribution is also unstable before that. Once the model weights are quantized, we see a regularization effect and the performance becomes much more stable on the OOD data. We provide training plots encompassing different domains as target settings for the sake of completeness. This inclusion serves to illustrate that quantization genuinely enhances stability in the training process. On the left, "te_location_100" is considered as target domain while "te_location_46" is used as the target domain for the plot on the right. These experiments evidence that model selection based on the in-domain validation set is much more reliable when introducing quantization into training.

### 3.5 ENSEMBLES OF QUANTIZATION

For our ensemble creation, we train multiple models independently from initialization, using random seeds to ensure diversity and, incorporate quantization into the training process to obtain smaller quantized models. We refer to this as the Ensemble of Quantization (EoQ). As Breiman (1996), we

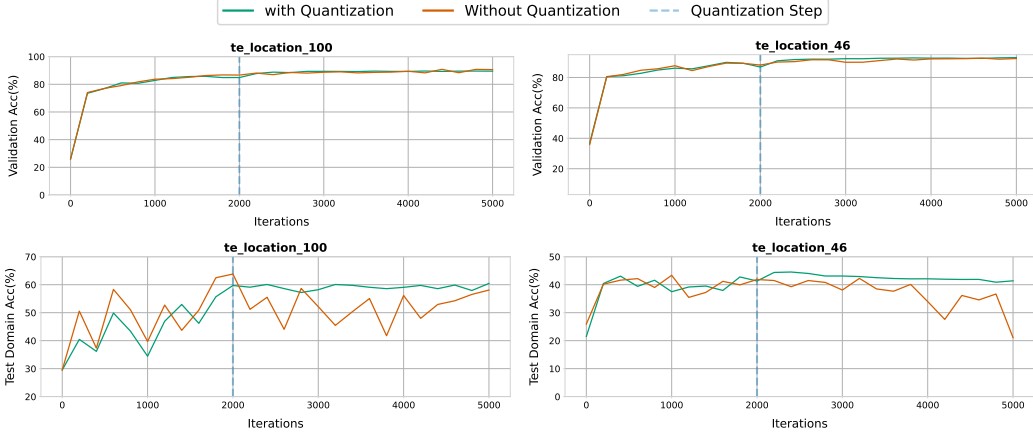

Figure 3: **Model Quantization improves out-of-domain performance as well as training** *stability*. The plots were computed using the TerraInc dataset with domain L100 (left) and L46 (right) as test domain, and the other domains as training/validation data. The top two plots illustrate in-domain validation accuracy, while the bottom two represent out-of-domain test accuracy. The network used for these plots was a ResNet-50. For our quantized models, shown in blue in each plot, we quantized the model after 2000 steps. Note that the model accuracy is not only better with quantization but also much more stable for out of distribution data after the quantization step.

use the bagging method to combine the multiple predictions. Therefore, the class predicted by EoQ for an input $\mathbf{x}$ is given by

$$\hat{y} = \arg \max_k \ \text{Softmax} \left( \frac{1}{E} \sum_{i=1}^{E} f(\mathbf{x}; \boldsymbol{w}_q^i) \right)_k \tag{6}$$

where $E$ is the total number of models in the ensemble, $\boldsymbol{w}_q^i$ denotes the parameters of the $i^{th}$ quantized model, and the subscript $k$ denotes the $k^{th}$ element of the vector argument. Finally, we use the in-domain validation set performance to pick the best model state (weights) $\boldsymbol{w}_q^i$ of the $i^{th}$ quantized model used in the ensemble.

## 4 EXPERIMENTS

### 4.1 DATASETS AND METRICS

We demonstrate the effectiveness of our proposed method on diverse classification datasets used for evaluating multi-source Domain Generalization:

**PACS** (Li et al., 2017) is a 7 object classification challenge encompassing four domains, with a total of 9,991 samples. It serves to validate our method in smaller-scale settings. **VLCS** (Fang et al., 2013) poses a 5 object classification problem across four domains. With 10,729 samples, VLCS provides a good benchmark for close Out-of-Distribution (OOD), featuring subtle distribution shifts simulating real-life scenarios. **OfficeHome** (Venkateswara et al., 2017) comprises a total of 15,588 samples. It presents a 65-way classification challenge featuring everyday objects across four domains. **TerraIncognita** (Beery et al., 2018) addresses a 10 object classification challenge of animals captured in wildlife cameras, with four domains representing different locations. The dataset contains 24,788 samples, illustrating a realistic use-case where generalization is crucial. **DomainNet** (Peng et al., 2019) provides a 345 object classification problem spanning six domains. With 586,575 samples, it is one of the largest datasets.

We report out-of-domain accuracies for each domain and their average, i.e., a model is trained and validated on training domains and evaluated on the unseen target domain. Each out-of-domain performance is an average of three different runs with different train-validation splits for the quantized models. We then combine the predictions of the different quantized models for our EoQ results.

### 4.2 IMPLEMENTATION DETAILS
All implementation details are provided in the Appendix.

### 4.3 RESULTS

In this section, we demonstrate the superior performance of our proposed approach by comparing it to recent state-of-the-art DG methods. We also present some visual evidence for the better performance of our quantization approach. Furthermore, we show how quantization not only enhances model generalization but also yields better performance on in-domain data.

Table 1: **Comparison with domain generalization methods.** Performance benchmarking on 5 datasets of the DomainBed benchmark. Highest accuracy is shown in bold, while second best is underlined. $^\dagger$ do not report confidence interval and ensembles do not have confidence interval because an ensemble uses all the models to make a prediction. Our proposed method is colored in Gray. Average accuracies and standard errors are reported from three trials. For all the reported results, we use the same training-domain validation protocol as (Gulrajani & Lopez-Paz, 2021). $Models$ corresponds to the number of models trained during training and $Size$ corresponds to the relative network size.

| Algorithm | Models | Size | PACS | VLCS | Office | TerraInc | DomainNet | Avg. |
|---|---|---|---|---|---|---|---|---|
| ResNet-50 (25M Parameters, Pre-trained on ImageNet) | | | | | | | | |
| ERM | 1 | 1x | $84.7 \pm 0.5$ | $77.4 \pm 0.3$ | $67.5 \pm 0.5$ | $46.2 \pm 0.4$ | $41.2 \pm 0.2$ | 63.8 |
| IRM | 1 | 1x | $84.4 \pm 1.1$ | $78.1 \pm 0.0$ | $66.6 \pm 1.0$ | $47.9 \pm 0.7$ | $35.7 \pm 1.9$ | 62.5 |
| Group DRO | 1 | 1x | $84.1 \pm 0.4$ | $77.2 \pm 0.6$ | $66.9 \pm 0.3$ | $47.0 \pm 0.3$ | $33.7 \pm 0.2$ | 61.8 |
| Mixup | 1 | 1x | $84.3 \pm 0.5$ | $77.7 \pm 0.4$ | $69.0 \pm 0.1$ | $48.9 \pm 0.8$ | $39.6 \pm 0.1$ | 63.9 |
| MLDG | 1 | 1x | $84.8 \pm 0.6$ | $77.1 \pm 0.4$ | $68.2 \pm 0.1$ | $46.1 \pm 0.8$ | $41.8 \pm 0.4$ | 63.6 |
| CORAL | 1 | 1x | $86.0 \pm 0.2$ | $77.7 \pm 0.5$ | $68.6 \pm 0.4$ | $46.4 \pm 0.8$ | $41.8 \pm 0.2$ | 64.1 |
| MMD | 1 | 1x | $85.0 \pm 0.2$ | $76.7 \pm 0.9$ | $67.7 \pm 0.1$ | $49.3 \pm 1.4$ | $39.4 \pm 0.8$ | 63.6 |
| Fish | 1 | 1x | $85.5 \pm 0.3$ | $77.8 \pm 0.3$ | $68.6 \pm 0.4$ | $45.1 \pm 1.3$ | $42.7 \pm 0.2$ | 63.9 |
| Fishr | 1 | 1x | $85.5 \pm 0.4$ | $77.8 \pm 0.1$ | $67.8 \pm 0.1$ | $47.4 \pm 1.6$ | $41.7 \pm 0.0$ | 65.7 |
| SWAD | 1 | 1x | $88.1 \pm 0.4$ | $\underline{79.1 \pm 0.4}$ | $70.6 \pm 0.3$ | $50.0 \pm 0.4$ | $46.5 \pm 0.2$ | 66.9 |
| MIRO | 1 | 1x | $85.4 \pm 0.4$ | $79.0 \pm 0.0$ | $70.5 \pm 0.4$ | $50.4 \pm 1.1$ | $44.3 \pm 0.2$ | 65.9 |
| CCFP | 1 | 1x | $86.6 \pm 0.2$ | $78.9 \pm 0.3$ | $68.9 \pm 0.1$ | $48.6 \pm 0.4$ | $41.2 \pm 0.0$ | 64.8 |
| ARM$^\dagger$ | 1 | 1x | 85.1 | 77.6 | 64.8 | 45.5 | 35.5 | 61.7 |
| VREx$^\dagger$ | 1 | 1x | 84.9 | 78.3 | 66.4 | 46.4 | 33.6 | 61.9 |
| RSC$^\dagger$ | 1 | 1x | 85.2 | 77.1 | 65.5 | 46.6 | 38.9 | 62.7 |
| Mixstyle$^\dagger$ | 1 | 1x | 85.2 | 77.9 | 60.4 | 44.0 | 34.0 | 60.3 |
| SagNet$^\dagger$ | 1 | 1x | 86.3 | 77.8 | 68.1 | 48.6 | 40.3 | 64.2 |
| QT-DoG (ours) | 1 | **0.22x** | $87.8 \pm 0.3$ | $78.4 \pm 0.4$ | $68.9 \pm 0.6$ | $50.8 \pm 0.2$ | $45.1 \pm 0.9$ | 66.2 |
| ERM Ens. $^\dagger$ | 6 | 6x | 87.6 | 78.5 | 70.8 | 49.2 | $\underline{47.7}$ | 66.8 |
| DiWA$^\dagger$ | 60 | 1x | $\underline{89.0}$ | 78.6 | 72.8 | 51.9 | $\underline{47.7}$ | $\underline{68.0}$ |
| EoA$^\dagger$ | 6 | 6x | 88.6 | $\underline{79.1}$ | **72.5** | $\underline{52.3}$ | 47.4 | $\underline{68.0}$ |
| DART | 4-6 | 4x-6x | $78.5 \pm 0.7$ | $87.3 \pm 0.5$ | $70.1 \pm 0.2$ | $48.7 \pm 0.8$ | $45.8 \pm 0.0$ | 66.1 |
| EoQ (ours)$^\dagger$ | 5 | 1.1x | **89.3** | **79.5** | $\underline{72.3}$ | **53.2** | **47.9** | **68.4** |

### 4.3.1 COMPARISON WITH DG METHODS

Table 1 reports out-of-domain performances on five DG benchmarks and compares our proposed approaches to prior works. These results demonstrate the superiority of EoQ across five DomainBed datasets, with an average improvement of 0.4% over the state-of-the-art EoA while reducing the memory footprint by approximately 75%. Compared to DiWA, we significantly reduce the computational burden and memory requirements for training, achieving a 12-fold reduction, as DiWA requires training 60 models for diverse averaging. EoQ achieves the most significant gain (7% improvement) on TerraIncognita (Beery et al., 2018), with nonetheless substantial gains of 3-5% w.r.t. ERM on PACS (Li et al., 2017) and DomainNet (Peng et al., 2019).

The results also demonstrate that simply introducing quantization into the ERM-based approach (Gulrajani & Lopez-Paz, 2021) surpasses or yields comparable accuracy to many existing works, although the size and computational budget of our quantization-based approach is significantly lower than that of the other methods. For our results in Table 1 and Figure 1, we employed 7-bit quantization on the network. Therefore, as shown in Figure 1, the model size is drastically reduced, becoming more than 4 times smaller than the other methods. Being smaller in memory footprint, our quantization-based approach can utilize ensembling without increasing the memory storage and computational resources. Moreover, quantization not only reduces the memory footprint but also the latency of the model. For example, running a ResNet-50 model on an **AMD EPYC 7302** processor yields a latency of 34.28ms for full-precision and 21.02ms for our INT8 quantized model.

### 4.3.2 COMBINATIONS WITH OTHER METHODS

Since QT-DoG requires no modifications to training procedures or model architectures, it is universally applicable and can seamlessly integrate with other DG methods. As shown in Table 3, we integrate QT-DoG with CORAL (Sun et al., 2016) and MixStyle (Zhou et al., 2021). Both CORAL and MixStyle demonstrate improved performance when combined with QT-DoG, reinforcing our findings that QAT aids in identifying flat minima, thereby enhancing domain generalization.

| Algorithm | Type | In-domain | Out-domain |
|---|---|---|---|
| No quant | - | $96.6 \pm 0.2$ | $84.7 \pm 0.5$ |
| OBC | PTQ | $96.8 \pm 0.2$ | $83.7 \pm 0.4$ |
| INQ | QAT | $97.1 \pm 0.2$ | $87.4 \pm 0.3$ |
| LSQ | QAT | $\mathbf{97.3 \pm 0.2}$ | $\mathbf{87.8 \pm 0.3}$ |

Table 2: **Model quantization with different quantization algorithms.** We report the average target domain accuracy and the average source domain accuracy across all domains in PACS.

| Algorithm | PACS | TerraInc | C |
|---|---|---|---|
| CORAL | $85.5 \pm 0.6$ | $47.1 \pm 0.2$ | - |
| CORAL + QT-DoG | $\mathbf{86.9 \pm 0.2}$ | $\mathbf{50.6 \pm 0.3}$ | 4.6x |
| MixStyle | $85.2 \pm 0.3$ | $44.0 \pm 0.4$ | - |
| MixStyle + QT-DoG | $\mathbf{86.8 \pm 0.3}$ | $\mathbf{47.7 \pm 0.2}$ | 4.6x |

Table 3: **Combination with other methods.** Results of PACS and Terra Incognita datasets incorporating QT-DoG with CORAL and MixStyle. C represents the compression factor of the model.

| Algorithm | Backbone | PACS | TerraInc | Compression |
|---|---|---|---|---|
| ERM_ViT | Deit-Small | $84.3 \pm 0.2$ | $43.2 \pm 0.2$ | - |
| ERM-SD_ViT | Deit-Small | $\mathbf{86.3 \pm 0.2}$ | $44.3 \pm 0.2$ | - |
| ERM_ViT + QT-DoG | Deit-Small | $86.2 \pm 0.3$ | $\mathbf{45.6 \pm 0.4}$ | **4.6x** |

Table 4: **Quantization of a Vision Transformer** Comparison of performance on PACS and TerraInc datasets with and without QT-DoG quantization of ERM_ViT (Sultana et al., 2022) with DeiT-Small backbone.

### 4.3.3 BIT PRECISION ANALYSIS

Here, we empirically analyze the effect of different bit-precisions for quantization on the generalization of the model. We perform experiments with four different bit levels and present an analysis in Figure 4 on the PACS (Li et al., 2017) and TerraIncognita (Beery et al., 2018) datasets. We report the test[1] domain accuracy averaged across all domains. For both datasets, 7-bit precision was found to be the optimal bit precision to have the best out-of-domain generalization while maintaining in-domain accuracy. Nonetheless, 8 bits and 6 bits also show improvements, albeit smaller than with 7-bit quantization. These results evidence that, even with a 6 times smaller model, quantization still yields better out-of-domain performance without sacrificing the in-domain accuracy.

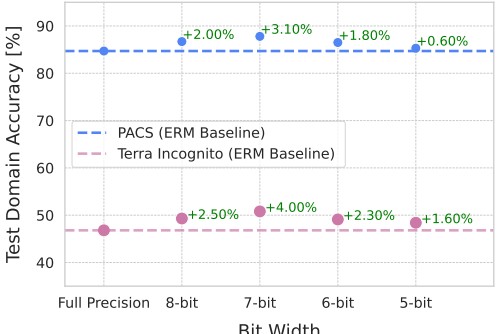

Figure 4: **Bit precision analysis for efficient quantization.** We show results on out-of-domain test accuracy with two different datasets, i.e., PACS and TerraIncognita. For each bit precision, we report the increase in the test domain accuracy averaged across all domains. The 7-bit quantized model exhibits the maximum increase for both datasets. We quantize the model at 2000 steps.

### 4.3.4 DIFFERENT QUANTIZATION METHODS

In this section, we perform an ablation study by replacing LSQ (Esser et al., 2020) with other quantization algorithms. We use INQ (Zhou et al., 2017) as another quantization-aware training method but also perform quantization using OBC (Frantar et al., 2022), that uses a more popular post-training quantization (PTQ) approach to quantize a network. We perform this ablation study on the PACS dataset, and the results are shown in Table 2. All the experiments are performed with 7-bit quantization. We observe that, while the QAT approaches tend to enhance generalization, the PTQ approach fails to do so. This is due to the fact that there is no training involved after the quantization step in PTQ. That is, with PTQ, we do not train the network with quantization noise to find a flatter minimum.

### 4.3.5 GENERALITY WITH VISION TRANSFORMER

In Table 4, we present the results of quantizing a vision transformer (ERM-ViT, DeiT-small) (Sultana et al., 2022) for domain generalization. We compare the performance of the baseline ERM-ViT to its quantized counterpart on the PACS and Terra Incognita datasets, demonstrating QT-DoG's effectiveness across different architectures. The results clearly show that QT-DoG also improves the performance of vision transformers. Additionally, we provide results for ResNeXt-50 32x4d in the appendix, following a similar evaluation as in Arpit et al. (2022).

---

[1]In-domain results for different bit-precisions are provided in the appendix.

Figure 5: **GradCAM visualization for ERM (Gulrajani & Lopez-Paz, 2021) and QT-DoG.** We show results on the PACS dataset (Li et al., 2017) and consider a different domain as test domain in each run, indicated by the different columns in the figure.

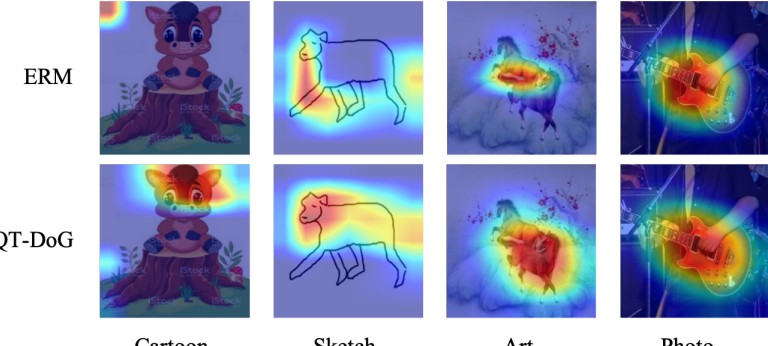

ERM

QT-DoG

Cartoon      Sketch      Art      Photo

### 4.3.6 VISUALIZATIONS

**GradCAM Results.** In Figure 5, we present some of the examples[2] from the PACS dataset and show GradCAM (Gildenblat & contributors, 2021) results in the target domain. We perform four different experiments by considering a different target domain for each run, while utilizing the other domains for training. We use the output from the last convolutional layer of the models with and without quantization. Both models are trained under the same settings as in Gulrajani & Lopez-Paz (2021). For our method, we quantize the model after 2000 iteration and employ 7-bit precision as it provides the best out-of-domain performance.

These visualizations evidence that quantization focuses on better regions than ERM, and with a much larger receptive field. In certain cases, ERM does not even focus on the correct image region. It is quite evident that quantization pushes the model to learn more generalized patterns, leading to a model that is less sensitive to the specific details of the training set. These qualitative results confirm the quantitative evidence provided in Table 1.

## 5 DISCUSSION AND LIMITATIONS

Despite showing success and surpassing the state-of-the-art methods in terms of performance, EoQ also has some limitations. First, it requires training multiple models like Rame et al. (2022b); Arpit et al. (2022), to create diversity and form an ensemble. This ensemble creation increases the training computational load. Nevertheless, our quantized ensembling models are much smaller in size.

Another limitation of this work is the challenge of determining the optimal bit precision for achieving the best performance in OOD generalization. In our experiments on the DomainBed benchmark, we identified 7 bits as the optimal precision. However, this may not hold true for other datasets. A potential future direction is to utilize a small number of target images to identify the optimal bit precision, which would significantly reduce the computational overhead associated with this process.

Lastly, given our utilization of a uniform quantization strategy, it would be interesting to investigate whether specific layers can be more effectively exploited than others through mixed-precision techniques to have even better domain generalization performance.

## 6 CONCLUSION

We introduced QT-DoG, a novel generalization strategy based on neural network quantization. Our approach leverages the insight that QAT can find flatter minima in the loss landscape, serving as an effective regularization method to reduce overfitting and enhance the generalization capabilities. We empirically demonstrated, supported by analytical insights, that quantization not only enhances generalization but also helps stabilize the training process. Our extensive experiments across diverse datasets show that incorporating quantization with an optimal bit-width significantly enhances domain generalization, yielding performance comparable to existing methods while reducing the model size. Additionally, we proposed EoQ, a powerful ensembling strategy that addresses the challenges of memory footprint and computational load by creating ensembles of quantized models. EoQ outperforms state-of-the-art methods while being approximately four times smaller than its full-precision ensembling counterparts.

---

[2]More examples are provided in the appendix.

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

## A    PER-DOMAIN PERFORMANCE IMPROVEMENT

We also report per-domain performance improvement for PACS (Li et al., 2017) and Terra Incognito (Beery et al., 2018) dataset. We choose the best model based on the validation set and report the results in 5 and 6. The results with quantization correspond to 7 bit-precision and we perform quantization after 2000 steps. Table 5 and 6 show that EoQ is consistently better than the current state-of-the-art methods across domains for different datasets.

| Algorithm | Art | Cartoon | Painting | Sketch | Avg. |
|---|---|---|---|---|---|
| ERM (our runs) | 89.8 | 79.7 | 96.8 | 72.5 | 84.7 |
| SWAD | 89.3 | 83.4 | 97.3 | 82.5 | 88.1 |
| EoA | 90.5 | 83.4 | 98.0 | 82.5 | 88.6 |
| DiWA | 90.6 | 83.4 | **98.2** | 83.8 | 89.0 |
| QT-DoG | 89.1 | 82.4 | 96.9 | 82.3 | 87.8 |
| EoQ | **90.7** | **83.7** | **98.2** | **84.8** | **89.3** |

Table 5: **Per-Domain Accuracy Comparison for PACS.** We report the accuracy for each domain of the PACS dataset along with the average across all domains. Our proposed quantization is shaded in Gray.

| Algorithm | L100 | L38 | L43 | L46 | Avg. |
|---|---|---|---|---|---|
| ERM (our runs) | 58.2 | 38.3 | 57.1 | 35.1 | 47.2 |
| SWAD | 55.4 | 44.9 | 59.7 | 39.9 | 50.0 |
| DiWA | 57.2 | 50.1 | 60.3 | 39.8 | 51.9 |
| EoA | 57.8 | 46.5 | **61.3** | 43.5 | 52.3 |
| QT-DoG | 60.2 | 46.4 | 55.2 | 41.4 | 50.8 |
| EoQ | **61.8** | **48.2** | 59.2 | **43.7** | **53.2** |

Table 6: **Per-Domain Accuracy Comparison for Terra Incognito.** We report the accuracy for each domain of the Terra Incognito dataset along with the average across all domains. Our proposed quantization is shaded in Gray.

## B    BIT PRECISION ANALYSIS EXTENDED

In contrast to main manuscript, Table 7 provides all the results in a tabular form. We show how quantization outperforms the vanilla ERM approach. This shows the superior performance of quantization over ERM despite being more than 6 times smaller in the case of 5 bit-precision.

| Algorithm | Compression | PACS | | TerraInc | |
|---|---|---|---|---|---|
| | | In-domain | Out-domain | In-domain | Out-domain |
| ERM (our runs) | - | $96.9 \pm 0.1$ | $84.7 \pm 0.5$ | $91.7 \pm 0.2$ | $47.2 \pm 0.4$ |
| QT-DoG(8) | 4x | $97.0 \pm 0.1$ | $85.0 \pm 0.1$ | $90.9 \pm 0.2$ | $49.1 \pm 0.1$ |
| QT-DoG(7) | 4.6x | $\mathbf{97.3 \pm 0.2}$ | $\mathbf{87.8 \pm 0.3}$ | $\mathbf{92.3 \pm 0.2}$ | $\mathbf{50.8 \pm 0.2}$ |
| QT-DoG(6) | 5.3x | $97.1 \pm 0.1$ | $86.5 \pm 0.1$ | $91.1 \pm 0.0$ | $49.0 \pm 0.3$ |
| QT-DoG(5) | 6.4x | $97.0 \pm 0.1$ | $85.3 \pm 0.4$ | $91.0 \pm 0.1$ | $48.4 \pm 0.2$ |

Table 7: **Model quantization with different bit-precisions vs vanilla ERM.** We report the average target domain accuracy as well as the average source domain accuracy across all domains for the PACS (Li et al., 2017) and TerraIncognita (Beery et al., 2018) datasets. Quantization not only enhances the generalization ability but also retains the source domain performance. QT-DoG(x) indicates a model quantized with x bit-precision.

However, as shown in Table 8, decreasing bit-precision through quantization does not always improve performance above the baseline; after a point, there is a tradeoff between compression and generalization. Specifically, our experiments with 4-bit precision and lower did not yield satisfactory results - see the Table below. Finding the sweet spot for balancing speed and performance can be an interesting research direction. Our results evidence that there exist configurations that can improve both speed and performance.

| Algorithm | Bit-Precision | PACS |
|---|---|---|
| ERM | 32 | $84.7 \pm 0.5$ |
| QT-DoG | 7 | $\mathbf{87.8 \pm 0.3}$ |
| | 6 | $86.5 \pm 0.1$ |
| | 5 | $85.3 \pm 0.4$ |
| | 4 | $84.3 \pm 0.3$ |
| | 3 | $83.3 \pm 0.4$ |
| | 2 | $82.8 \pm 0.2$ |

Table 8: **Effect of aggressive quantization.** Performance comparison between ERM and QT-DoG with varying bit-precision on PACS.

## C  EXPERIMENTS WITH LARGER PRE-TRAINING DATASETS

We also show experimental results with ResNeXt-50-32x4 in Table 9. Note that both ResNet-50 and ResNeXt-50-32x4d have 25M parameters. However, ResNeXt-50-32x4d is pre-trained on a larger dataset i.e Instagram 1B images(Yalniz et al., 2019). It is evident from Table 9 that incorporating quantization into training consistenlty improve accuracy even when a network is pre-trained on a larger dataset. Furthermore, EoQ again showed superior performance in comparison to other methods across five DomainBed datasets.

| Algorithm | M | S | PACS | VLCS | Office | TerraInc | DomainNet | Avg. |
|---|---|---|---|---|---|---|---|---|
| | | | ResNeXt-50 32x4d (25M Parameters, Pre-trained 1B Images) | | | | | |
| ERM | 1 | 1x | $88.7 \pm 0.3$ | $79.0 \pm 0.1$ | $70.9 \pm 0.5$ | $51.4 \pm 1.2$ | $48.1 \pm 0.2$ | 67.7 |
| SMA | 1 | 1x | $92.7 \pm 0.3$ | $79.7 \pm 0.3$ | $78.6 \pm 0.1$ | $53.3 \pm 0.1$ | $53.5 \pm 0.1$ | 71.6 |
| QT-DoG (ours) | 1 | 1x | $92.9 \pm 0.3$ | $79.2 \pm 0.4$ | $78.9 \pm 0.3$ | $54.1 \pm 0.2$ | $53.9 \pm 0.2$ | 71.8 |
| ERM Ens.[†] | 6 | 6x | 91.2 | 80.3 | 77.8 | 53.5 | 52.8 | 71.1 |
| EoA[†] | 6 | 6x | 93.2 | **80.4** | 80.2 | 55.2 | 54.6 | 72.7 |
| EoQ[†] (ours) | **5** | **1.1x** | **93.5** | 80.3 | **80.3** | **55.6** | **54.8** | **72.9** |

Table 9: **Comparison with other methods for ResNeXt-50.** Performance benchmarking on 5 datasets of the DomainBed benchmark. Highest accuracy is shown in bold, while second best is underlined. Ensembles[†] do not have confidence interval because an ensemble uses all the models to make a prediction. Our proposed method is colored in Gray. Average accuracies and standard errors are reported from three trials. For all the reported results, we use the same training-domain validation protocol as (Gulrajani & Lopez-Paz, 2021). $M$ corresponds to the number of models trained during training and $S$ corresponds to the relative network size.

## D  IN-DOMAIN PERFORMANCE IMPROVEMENT USING QUANTIZATION

We further study the in-domain test accuracy of our quantization approach without ensembling on PACS and TerraIncognita datasets. As (Cha et al., 2021), we split the in-domain datasets into training (60%), validation (20%), and test (20%) sets. We choose the best model based on the validation set and report the results on the test set in Table 10. The results with quantization correspond to 7 bit-precision.

QT-DoG also enhances the in-domain performance. The regularization effect introduced by quantization prevents the model from overfitting to edge cases and pushes it to learn more meaningful and generalizable features, which we also demonstrate in Section 4.3.6. As the training data consists of various domains and the quantization limits the range of weight values, it discourages the model from becoming overly complex and overfitting to the noise in the training data. Therefore, the model is more robust to minor input fluctuations.

| Method | PACS | TerraInc | Compression |
|--------|------|----------|-------------|
| ERM | 96.6 ± 0.2 | 90.1 ± 0.2 | - |
| SAM | 97.3 ± 0.1 | 90.8 ± 0.1 | - |
| SWA | 97.1 ± 0.1 | 90.7 ± 0.1 | - |
| SMA | 96.8 ± 0.2 | 90.7 ± 0.4 | - |
| SWAD | **97.7 ± 0.2** | 90.8 ± 0.3 | - |
| QT-DoG | 97.3 ± 0.2 | **91.1 ± 0.2** | ∼4.6x |

Table 10: **Comparison between generalization methods on PACS and TerraInc for IID settings.** We report the accuracy averaged across all domains. Our proposed approach is shaded in Gray. Highest accuracy is shown in bold, while second best is underlined.

# E    RESULTS ON WILDS DATASET

We performed experiments with 7 bit quantization on two datasets from the WILDS benchmark (Koh et al., 2021b). We utilized the same experimental settings as outlined in the WILDS benchmark repository and incorporated quantization into the training process. The results presented below confirm our findings on Domainbed [PACS, Terra, VLCS, Office, DomainNet] benchmark:

| Dataset | Method | In-dist | Out-dist | Metric |
|---------|--------|---------|----------|--------|
| Amazon | ERM | 71.9 ± 0.1 | 53.8 ± 0.8 | 10th percentile acc |
| Amazon | QT-DoG | **79.2 ± 0.5** | **55.9 ± 0.6** | 10th percentile acc |
| Camelyon | ERM | 93.2 ± 5.2 | 70.3 ± 6.4 | Average acc |
| Camelyon | QT-DoG | **96.4 ± 2.1** | **78.4 ± 2.2** | Average acc |

Table 11: **Comparison between ERM and QT-DoG on the Amazon and Camelyon datasets.** We report the in-domain and out-of-domain accuracy with respective metrics as shown.

# F    ABLATION ON LAYERWISE AND CHANNELWISE SCALE

we conducted an ablation study where we set $s$ at the layer level, rather than on a per-channel basis. We see that Channelwise $s$ can lead to 1.5% accuracy as compared to layerwise $s$. The results of this experiment on the PACS dataset with 7 bit quantization are shown below:

| Scale | OOD Accuracy |
|-------|--------------|
| No quantization | 84.7 ± 0.5 |
| Layerwise | 86.3 ± 0.4 |
| Channelwise | 87.8 ± 0.3 |

Table 12: OOD Accuracy with channelwise vs layerwise Scaling factor for quantization.

## G   ABLATION ON QUANTIZATION STEPS

We conducted an ablation study on the PACS dataset to identify the optimal number of steps after which quantization should be applied. We perform 7-bit quantization and the results are summarized below:

| Quantization Step | OOD Accuracy |
|---|---|
| No quantization | 84.7 ± 0.5 |
| 1000 | 86.2 ± 0.4 |
| 2000 | 87.8 ± 0.3 |
| 3000 | 86.9 ± 0.4 |
| 4000 | 85.1 ± 0.3 |

Table 13: OOD Accuracy across different quantization steps.

## H   VISUALIZATION

### H.1   MORE GRADCAM RESULTS

In Figure 7, 8, 9, 10, we present some of the examples from the Terra dataset and show Grad-CAM (Gildenblat & contributors, 2021) results on the target domain. We use the output from the last convolutional layer of the models with and without quantization for GradCAM. Similar to our experiments on PACS dataset, we perform four different experiments by considering a different target domain for each run, while utilizing the other domains for training. Both models are trained with the similar settings as (Gulrajani & Lopez-Paz, 2021). For quantization method, we quantized the model after 2000 iteration and employ 7 bit-precision as it provides the best out-of-domain performance. Moreover, we present some more examples for PACS dataset in Figure 6

These visualizations further proves that quantization pushes the model to be less sensitive to the specific details of the training set.

## I   IMPLEMENTATION DETAILS

We use the same training procedure as DomainBed (Gulrajani & Lopez-Paz, 2021), incorporating additional components from quantization. Specifically, we adopt the default hyperparameters from DomainBed (Gulrajani & Lopez-Paz, 2021), including a batch size of 32 (per-domain). We employ a ResNet-50 (He et al., 2016) pre-trained on ImageNet (Russakovsky et al., 2015) as initial model and use a learning rate of 5e-5 along with the Adam optimizer, and no weight decay. Following SWAD(Cha et al., 2021), the models are trained for 15,000 steps on DomainNet and 5,000 steps on the other datasets. In the training process, we keep a specific domain as the target domain, while the remaining domains are utilized as source domains. During this training phase, 20% of the samples are used for validation and model selection. We validate the model every 300 steps using held-out data from the source domains, and assess the final performance on the excluded domain (target).

We use LSQ (Esser et al., 2020) and INQ (Zhou et al., 2017) for model quantization, with the same configuration as existing quantization methods (Esser et al., 2020; Bhalgat et al., 2020; Dong et al., 2019; Yao et al., 2020; Zhou et al., 2017), where all layers are quantized to lower bit precision except the last one. We quantize the models at 8,000 steps for DomainNet and 2,000 steps for the other datasets. Moreover, each channel in a layer has a different scaling factor.

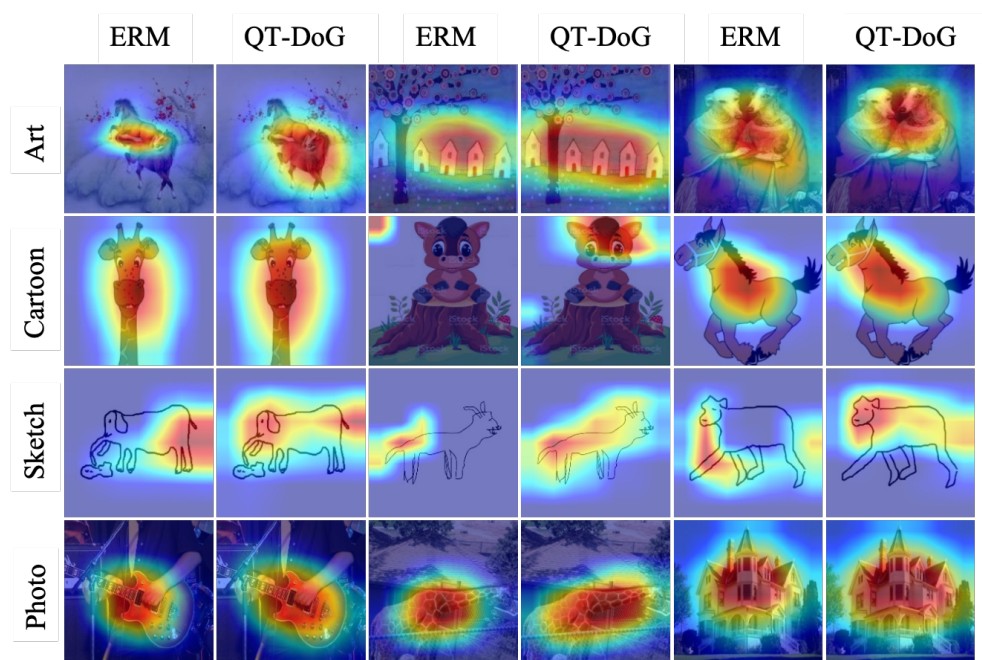

Figure 6: **GradCAM visualization for ERM (Gulrajani & Lopez-Paz, 2021) and QT-DoG.** We show results on the PACS dataset (Li et al., 2017) and consider a different domain as test domain in each run, indicated by the different rows in the figure.

## J REPRODUCIBILITY

To guarantee reproducibility, we will provide the source code publicly along with the details of the environments and dependencies. We will also provide instructions to reproduce the main results of Table 1 in the main paper. Furthermore, we will also share instructions and code to plot the loss surfaces and GradCAM results.

Every experiment in our work was executed on a single NVIDIA A100, Python 3.8.16, PyTorch 1.10.0, Torchvision 0.11.0, and CUDA 11.1.

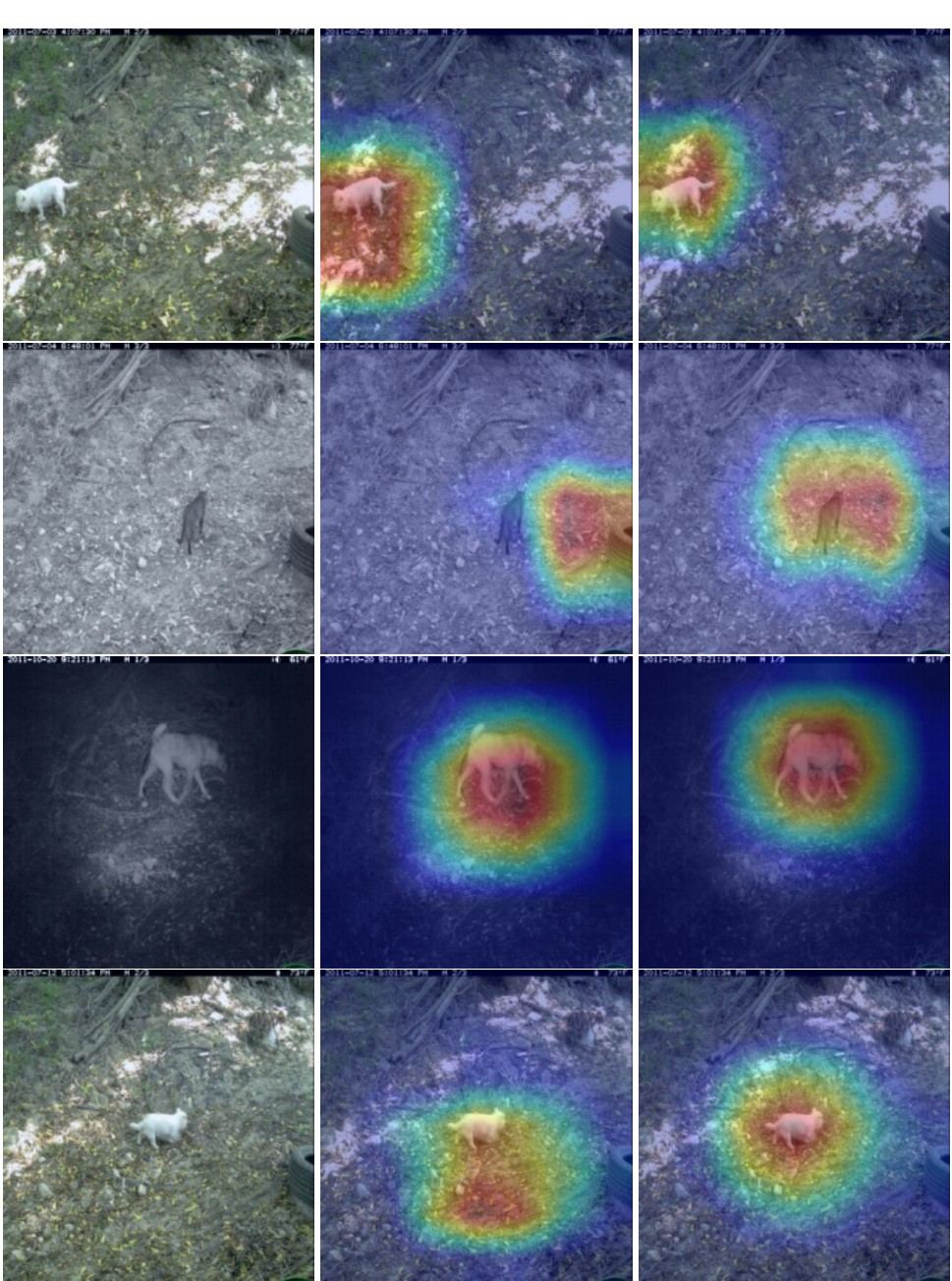

Figure 7: **Visualization of GradCAM results on the Terra Incognito dataset with L38 as test domain.** We show original image, GradCAM with ERM (Gulrajani & Lopez-Paz, 2021) and GradCAM with QT-DoG [Left to Right].

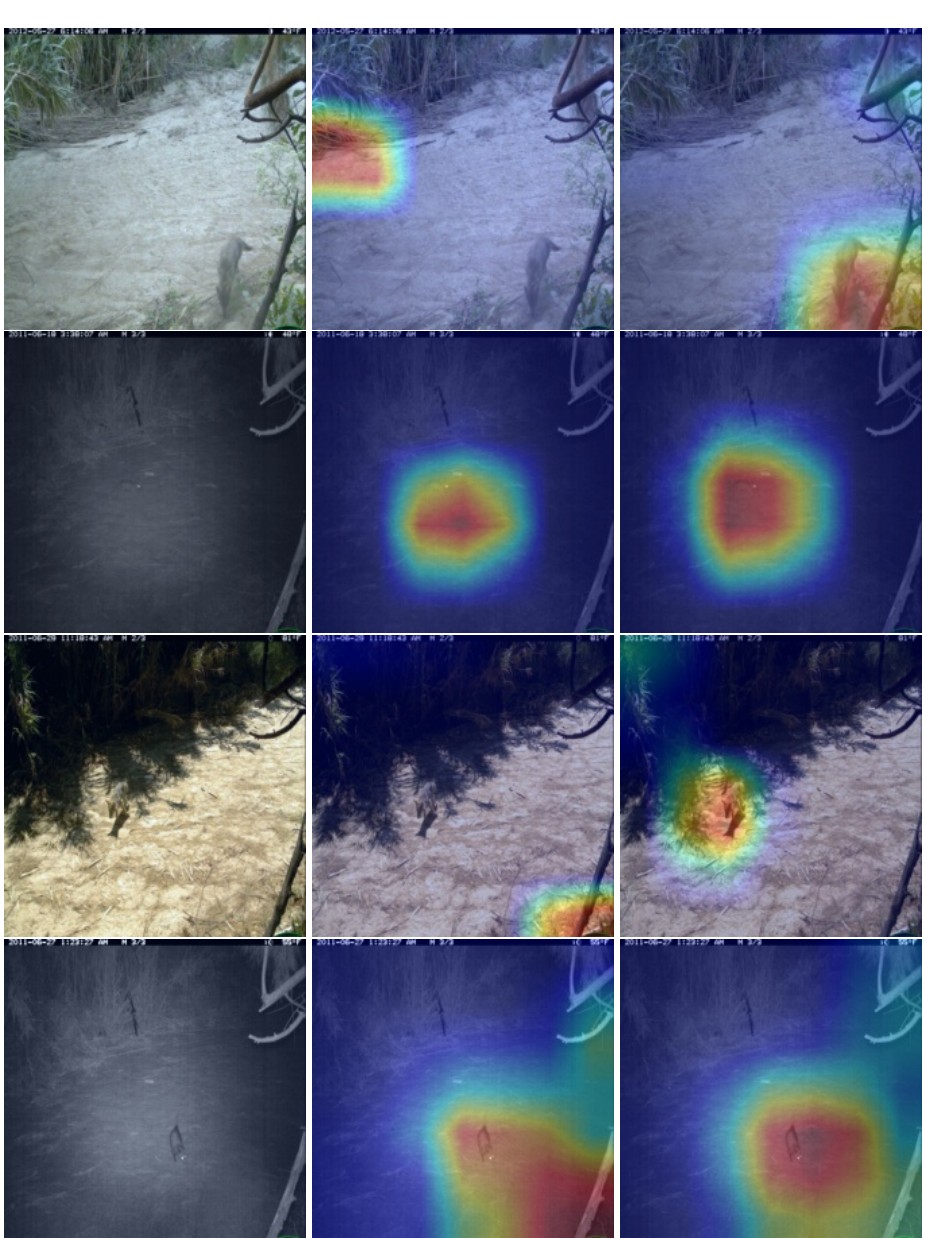

Figure 8: **Visualization of GradCAM results on the Terra Incognito dataset with L46 as test domain.** We show original image, GradCAM with ERM (Gulrajani & Lopez-Paz, 2021) and GradCAM with QT-DoG [Left to Right].

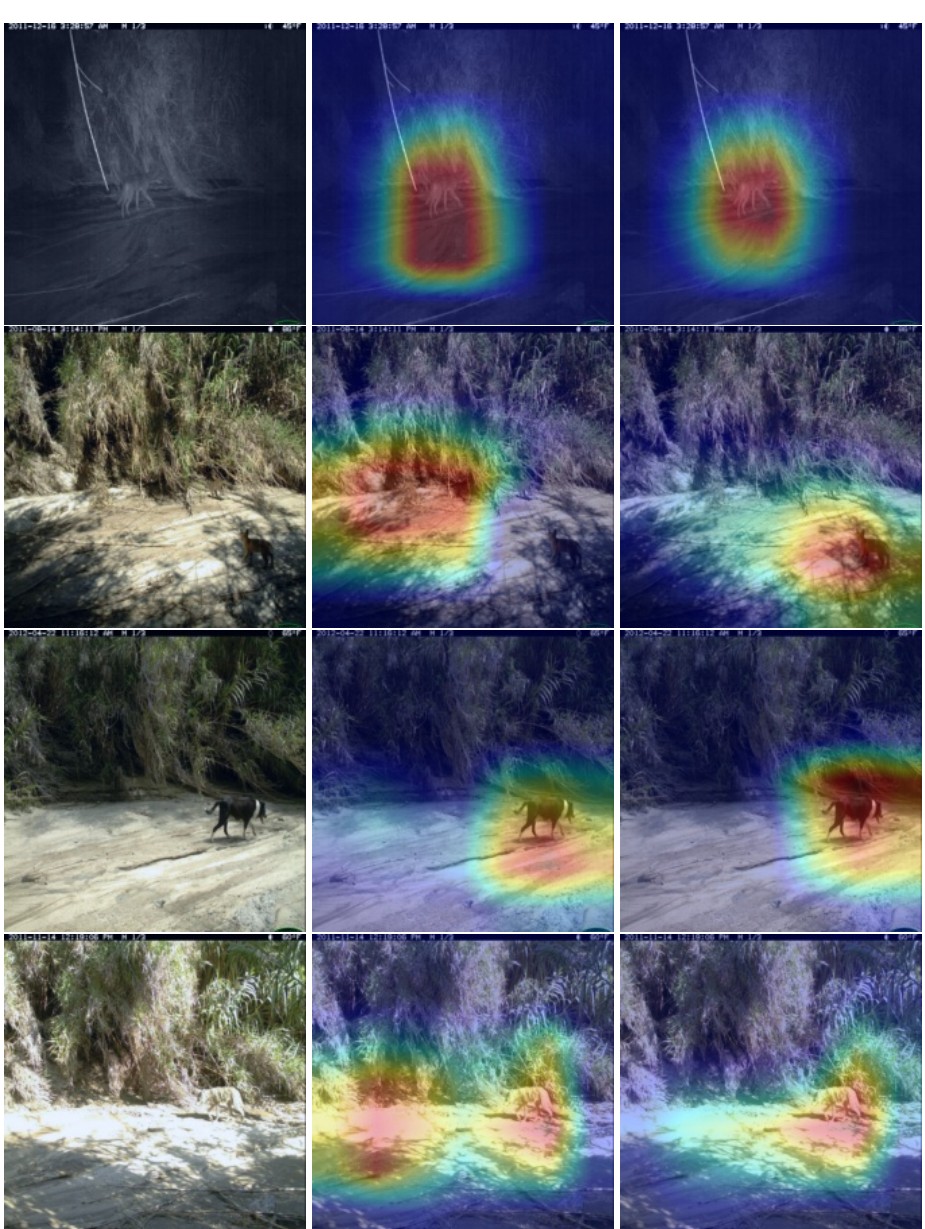

Figure 9: **Visualization of GradCAM results on the Terra Incognito dataset with L43 as test domain.** We show original image, GradCAM with ERM (Gulrajani & Lopez-Paz, 2021) and GradCAM with QT-DoG [Left to Right].

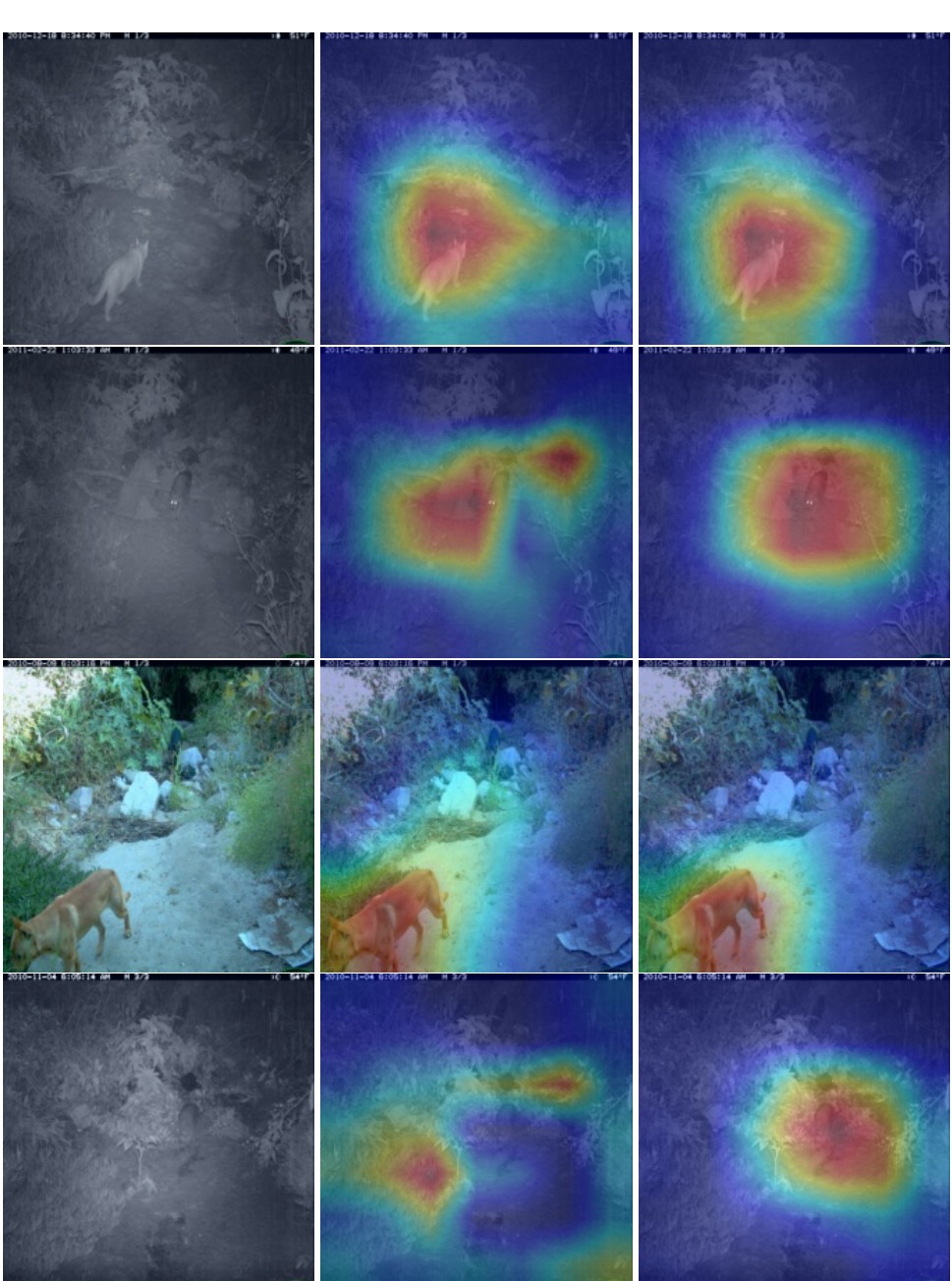

Figure 10: **Visualization of GradCAM results on the Terra Incognito dataset with L100 as test domain.** We show original image, GradCAM with ERM (Gulrajani & Lopez-Paz, 2021) and GradCAM with QT-DoG [Left to Right].

