# OpenReview forum: "QT-DoG: Quantization-Aware Training for Domain Generalization"
_ICLR.cc/2025/Conference — Submitted to ICLR 2025_

### Official Review · Reviewer_Rbwh · 2024-10-28

**Soundness:** 2
**Presentation:** 3
**Contribution:** 2
**Rating:** 3
**Confidence:** 4

**Summary:**

This paper proposes using quantization during model training as a strategy to enhance domain generalization. The authors demonstrate that the baseline ERM achieves competitive results when quantization is applied as an implicit regularizer, with quantization-induced noise guiding the model toward flatter minima in the loss landscape. Additionally, they introduce combining quantized models into a model soup, to further boost DG performance. Results on the DomainBed benchmark indicate that this approach performs comparably to state-of-the-art DG methods.

**Strengths:**

1. The paper is well-written, experimental results seem comparable against state-of-the-arts.

2. The design of using quantized models seem to be a good fit in weight ensembling, which has been proved effective for improving generalization.

**Weaknesses:**

1. The primary issue is the limited contribution. Their main idea is to replace the standard training in ERM with the existing quantized technique. That is hardly a new practice, since quantization itself is originally applied in ERM (given ERM is the basis of all training tasks), and they do not inroduce a DG-specific quantization technique. Additionally, the ensemble of qunatitized ERM is also an simple extension of the original model-soup. These can be called a trick for improving DG, but shouldn't be listed as a main contribution.

2. The overclaim is another problem. It is claimed that the theoretical connection between quantization and flatter minima is provided. To begin with, I cannot find a seriours definition for the flatter minima of a model in the manuscript. Moreover, even with the informal definition in Eq. (5), they do not provide any theoretical evidence to support that $\mathcal{F}\_{\gamma}(W^q) < \mathcal{F}\_{\gamma}(W)$. At last, I doubt it can be theoretically proved, given there are many quantization methods with different settings, it's hard to conclude a general framework to be applied in all.

3. According to Tab. 2, it seems different quantization methods can significantly affect the in-domain and out-of-domain accuracy, some even worse than the baseline ERM. Does this suggests that not all quantization leads to flat minima? If so, why does the adopted quantization method can lead to flat minima?

4. Experiments should be conducted in more data. The more realsitic WILDS dataset could be used to further show their effectiveness. Ablation studies shoube be conducted in more datasets rather than just PACS and TerraInc. Especially in Figure 3, where the upper two figures can barely support the claim.

5. Some unclear experimental settings. For example, how to choose the quantizer step size $s$? what is the quantizer step in Figure 3, is it a same thing with $s$. If no, how to decide it? does the quantization only applied in saving the model?

**Questions:**

See weakness

---

> ### Author Response · Authors · 2024-11-21
>
> We appreciate your recognition of our results' competitiveness and the efficiency of combining quantized models in ensembling. We have carefully considered your comments and have provided detailed responses to your queries below.
>
> >**W1: How does your approach of replacing standard ERM training with quantization and using an ensemble of quantized models offer a novel contribution to domain generalization, given that quantization and model-soup are already established techniques?**
>
>  While it is true that quantization has been applied in ERM in prior work, our key contribution lies in repurposing quantization as an implicit regularizer specifically for domain generalization (DG). Unlike traditional uses of quantization for compression, we show that quantization noise helps guide optimization toward flatter minima, which enhances OOD generalization. For more clarity, we reworded our first contribution as ``We are the first to demonstrate that quantization-aware training, traditionally used for model compression, can serve as an implicit regularizer, with quantization noise enhancing domain generalization."
>
> Moreover, our ensemble of quantized models builds on model-soup but introduces a key innovation by combining quantization with ensembling in a way that reduces resource overhead while maintaining high performance. This fusion not only improves DG but also ensures scalability across diverse benchmarks. The combination of these techniques within the context of domain generalization, supported by extensive empirical validation, represents a solid and impactful contribution, which we believe to be of interest to the community.
>
> >**W2: How does the manuscript define flat minima, and what theoretical evidence supports the connection between quantization and flatter minima?**
>
> We appreciate the reviewer’s feedback and acknowledge that the definition of flat minima needs to be more clearly presented. In the revised manuscript, we have clarified the discussion of flat minima. Flat minima refer to regions in the loss landscape where the loss remains relatively stable under small perturbations of the model parameters.
>
> The theoretical connection is drawn in Equation 4, which relates the Hessian of the loss to the impact of quantization. Quantization noise acts as a perturbation, and the optimization process biases the model toward flatter regions where the Hessian eigenvalues are smaller, ensuring stability under these perturbations. We have revised the text to clarify this.
>
> >**W3: Equation 5 does not provide theoretical evidence to support $\mathcal{F_\gamma(W^q)<F_\gamma(W)}$. How do you justify its inclusion, given the variety of quantization methods and settings?**
>
>
> Equation 5 is not intended to provide a theoretical proof or establish a formal connection but rather to serve as an empirical demonstration, akin to the approach used in SWAD [1]. It illustrates that the loss landscape becomes flatter with the application of quantization. This observation is supported by the plots in Figure 2, where it is evident that quantized models consistently achieve flatter loss landscapes compared to their non-quantized counterparts. While the diversity of quantization methods and settings makes a universal theoretical framework challenging, our empirical findings consistently validate this behavior across the datasets and quantization methods we evaluated.
>
> [1] SWAD: Domain generalization by seeking flat minima. Cha et al., NeurIPS 2021.
>
> >**W4: Not all quantization methods lead to flat minima. Why does the adopted method succeed where others fail?**
>
>
> This observation aligns with our findings. We demonstrate that quantization-aware training (QAT) encourages a flatter minima, which is not guaranteed with post-training quantization (PTQ). In QAT, flatter minima is achieved by incorporating quantization noise during training, acting as an implicit regularizer and smoothing the loss landscape (see Equation 4). In contrast, PTQ primarily involves clipping the network weights without any subsequent retraining., so it does not bring the same effect. This difference is reflected in Table 2, where PTQ performs worse on out-of-domain data compared to QAT. We provide a detailed discussion of this distinction in Section 4.3.4. If further clarification is needed, we are happy to revise this explanation.

---

> > ### Author Response · Authors · 2024-11-21
> >
> > >**W5: Include more realistic datasets like WILDS, to further demonstrate the effectiveness of your method**
> >
> > Following your suggestion, we performed experiments with 7 bit quantization on two datasets from the WILDS benchmark. Due to time limitations, we were unable to test all possible configurations. However, if there is a specific dataset that you believe would significantly impact the results or your perspective, we can prioritize running those experiments. We will also include additional experiments on the WILDS dataset in the revised manuscript to further substantiate our findings. We utilized the same experimental settings as outlined in the WILDS benchmark repository and incorporated quantization into the training process. The results presented below confirm our findings on Domainbed [PACS, Terra, VLCS, Office, DomainNet] benchmark:
> >
> > | Dataset| Method| In-dist |OOD |Metric|
> > |-|--|-|-|-|
> > | Amazon| ERM | 71.9 (0.1) | 53.8 (0.8)  | 10th percentile acc |
> > | Amazon| QT-DoG  | 79.2 (0.5)   | 55.9 (0.6)   |10th percentile acc |
> > | Camelyon| ERM| 93.2 (5.2)| 70.3 (6.4) |Average acc|
> > | Camelyon| QT-DoG| 96.4 (2.1) | 78.4 (2.2)  |Average acc|
> >
> >
> > >**W6: Why were the ablation studies conducted only on PACS and TerraInc, and how do you justify the claims in Figure 3, where the upper two figures may not seem sufficient to support your argument?**
> >
> > Ablation studies are commonly conducted on a single dataset; however, we have provided results on two datasets for greater robustness. We specifically chose PACS and TerraInc to ensure a balanced evaluation, with one larger and one smaller dataset represented. If there is a specific dataset you would like us to include for further analysis, please let us know, and we will be happy to provide additional results.
> >
> > Regarding Fig. 3, there appears to be some misunderstanding. The upper two figures display in-domain validation accuracy plots. The intention behind showing these is to demonstrate that the in-domain validation accuracy remains high even after quantization. Additionally, the bottom plots show that the out-of-domain test accuracy is not only higher but also more stable for the quantized model compared to the non-quantized model. We have revised the caption for more clarity.
> >
> > >**Q1: how to choose the quantizer step size $s$? Is it a same thing as quantization step in Fig. 3?**
> >
> > We opted for a learnable $s$ as it is considered best practice [1,2,3,4] in the field, allowing the model to adapt and optimize this parameter during training.
> >
> > In Figure 3, the ``quantization step" refers to the specific iteration at which quantization is applied during training.
> > Here are the results from our ablation study conducted on PACS dataset with 7-bit quantization:
> >
> > | Quantization Step | OOD Acc. |
> > |-|-|
> > | No quantization| 84.7 ± 0.5|
> > | 1000| 86.2 ± 0.4|
> > | 2000 | 87.8 ± 0.3|
> > | 3000 | 86.9 ± 0.4|
> > | 4000 | 85.1 ± 0.3|
> >
> > We have included these results in the appendix for further reference.
> >
> > [1] Mixed-Precision Neural Network Quantization via Learned Layer-wise Importance, ECCV 2022. \
> > [2] Learnable Companding Quantization for Accurate Low-bit Neural Networks, CVPR 2021. \
> > [3] Learned Step Size Quantization, ICLR 2020. \
> > [4] LSQ+: Improving low-bit quantization through learnable offsets and better initialization, CVPRW 2020.
> >
> > >**Q2: Does the quantization only applied in saving the model?**
> >
> > To clarify, we employ quantization-aware training (QAT), where quantization is applied after a certain number of training steps, and the model is subsequently trained with quantized weights. This approach results in quantized models that are not only smaller and faster but also exhibit enhanced generalization capabilities.

---

> > > ### Author Response · Authors · 2024-11-26
> > >
> > > We sincerely hope that our response has addressed your concerns and provided greater clarity. Your feedback is extremely valuable to us, and we would greatly appreciate it if you could kindly confirm whether we have adequately addressed your points. If you have any remaining questions or further concerns, we would be more than willing to provide additional clarification. Thank you once again for your time and feedback in helping us improve our work.

---

> > > > ### Comment · Reviewer_Rbwh · 2024-11-27
> > > > **Acknowledge the response**
> > > >
> > > > Thanks the author for the response. After checking the revised manuscript, my main concern remains.
> > > >
> > > > 1. I don't think there is much contribution for one to combine several previous techniques into an existing apllication without decent analysis.
> > > >
> > > > 2. I'm also not convinced by the claim of theoretical connection between the flat minima and quantization. Flat minima have been formally defined in previous work [a], with either Definitions 1 or 2 providing a clearer explanation than the current text. To support the theoretical claim, the authors need to provide serious proofs based on the definition (rather than intuitive explanations) to show that quantization leads to flatter minima. The current form with a simple Taylor expansion is too weak for such a strong claim.
> > > >
> > > > For the above reasons, I decide to maintain my rating.
> > > >
> > > > [a] Sharp Minima Can Generalize For Deep Nets, in ICML'17

---

> > > > > ### Author Response · Authors · 2024-11-28
> > > > >
> > > > > Thank you for your thoughtful comments. We would like to address the concerns you raised.
> > > > >
> > > > > >**Combining several previous techniques into an existing application without decent analysis**
> > > > >
> > > > > We acknowledge your perspective on our work. However, here’s why we think it should be accepted:
> > > > >
> > > > > 1) Quantization is popular, but no one has explored its use for boosting domain generalization performance.
> > > > > 2) We explain why it works (flatter minima), how it works (adding noise = regularization), and how to achieve it (use QAT, not PTQ).
> > > > > 3) We deliver state-of-the-art domain generalization models without any bells and whistles.
> > > > >
> > > > > >**Overclaim on the theoretical evidence since there is no formal proof.**
> > > > >
> > > > > The reviewer is correct that we do not provide a formal theoretical proof. As also noted in your original review, there may not be a formal proof that fits all existing methods. We appreciate this feedback and have adjusted our claims to better align with the evidence presented in the paper. Specifically:
> > > > >
> > > > > 1) In the contributions section, we revised the second claim from “We theoretically and empirically demonstrate that QAT encourages flatter minima” to “We empirically demonstrate that QAT promotes flatter minima in the loss landscape and provide an analytical perspective to explain this effect.”
> > > > > 2) In Section 3.3, we discuss the flatness definition mentioned in Hochreiter & Schmidhuber (1997), which links to both, Definition 1 in [a] and Cha et al. (2021). More specifically, we added “Similar to (Dinh et al., 2017; Cha et al., 2021), we interpret flat minima as “a large connected region in weight space where the error remains approximately constant,” as defined by (Hochreiter & Schmidhuber, 1997)“.
> > > > > 3) We modified “theoretical insights” to “analytical perspective” in the abstract, introduction, and conclusion.
> > > > >
> > > > > [a] Sharp Minima Can Generalize For Deep Nets, in ICML'17
> > > > >
> > > > > We sincerely thank the reviewer for this valuable feedback. We would greatly appreciate if the reviewer could clarify:
> > > > > 1) Whether our revised language resolves the issue of overclaiming.
> > > > > 2) If the three points we presented above are adequately justified in the paper.
> > > > > 3) If other concerns from your original review have been adequately addressed.
> > > > > 4) If there are other concerns or any particular analysis that would change your opinion on the paper.

---

> > > > > > ### Author Response · Authors · 2024-12-02
> > > > > > **Thank you!**
> > > > > >
> > > > > > Dear reviewer,
> > > > > > As the discussion period nears its end, we kindly hope to receive your feedback today. We have carefully highlighted all revisions in red and would greatly appreciate your kind consideration before finalizing your recommendation.
> > > > > >
> > > > > > We greatly value your input and hope the updates meet your expectations.

---

### Official Review · Reviewer_SqJY · 2024-11-04

**Soundness:** 3
**Presentation:** 3
**Contribution:** 3
**Rating:** 6
**Confidence:** 4

**Summary:**

This paper proposes quantization-aware training to improve models' domain generalization performance. Theoretical and empirical analysis show that quantization-aware training induces noise in model weights, which could guide the optimization process toward flatter minima that generalize better.

**Strengths:**

1. Introducing quantization to domain generalization, and drawing the potential link between quantization and flat minima is novel and helpful, as it could improve domain generalization performance as well as memory and computation efficiency.
2. Extensive experiments on DomainBed and empirical analysis show the effectiveness of QT-DoG.

**Weaknesses:**

1. The effect of $s$ should be discussed in more detail, as it plays an important role in the training and the theoretical analysis. Specifically, choosing a different $s$ of each channel in a layer should be justified. Ablation studies regarding $s$ can also be conducted to give a better understanding of the effect of $s$.

3. Measuring sharpness by equation 5 needs more discussion. When the model weight under measurement has different scales, perturbing weight with the same $\gamma$ does not serve as a fair way to get $w'$ around the local area of $w$. Relative flatness (Definition 3 in [1]) could be considered to address the dependency of sharpness measurement over scale of model weight.

[1] https://arxiv.org/pdf/2001.00939

**Questions:**

1. How much does $w_q$ differ from $w$ in training?
2. Why each channel in a layer has a different scaling factor? How is the value of $s$ determined?
3. Given a fixed quantization bit $b$, how would $s$ influence the performance?
4. What is the scale of $w$ (e.g. norm) across different algorithms in section 3.3?

---

> ### Author Response · Authors · 2024-11-21
>
> Thank you for acknowledging the novelty of using quantization to achieve flatter minima and the robustness of our experimental results. We have addressed your questions and concerns below:
>
> >**W1/Q2: Why each channel in a layer has a different scaling factor? How is the value of $s$ determined?**
>
> It is common practice[1,2,3] to use a learnable, channel-wise scaling factor $s$ because channels within a layer often exhibit varying activation and weight distributions. To account for these differences, channel-wise scaling factors are applied to normalize the perturbation $\Delta$ per channel. However, to explore the impact of this choice, we conducted an ablation study where we set $s$ at the layer level, rather than on a per-channel basis. We see that Channelwise $s$ can lead to 1.5% accuracy as compared to layerwise $s$. The results of this experiment on the PACS dataset with 7 bit quantization are shown below:
>
> | scale        | OOD Accuracy      |
> |-------------|-------------------|
> | No quantization    | 84.7 ± 0.5        |
> | Channelwise | 87.8 ± 0.3        |
> | Layerwise   | 86.3 ± 0.5        |
>
> [1] Mixed-Precision Neural Network Quantization via Learned Layer-wise Importance, ECCV 2022. \
> [2] Learnable Companding Quantization for Accurate Low-bit Neural Networks, CVPR 2021. \
> [3] Learned Step Size Quantization, ICLR 2020. \
> [4] LSQ+: Improving low-bit quantization through learnable offsets and better initialization, CVPRW 2020.
>
> >**W2: Measuring sharpness by equation 5 needs more discussion. When the model weight under measurement has different scales, perturbing weight with the same does not serve as a fair way to get around the local area of w'. Relative flatness (Definition 3 in [1]) could be considered to address the dependency of sharpness measurement over scale of model weight. [1] https://arxiv.org/pdf/2001.00939**
>
> Thank you for sugggesting this. We find this idea compelling and would like to conduct a proof of concept to explore its potential in our analysis. We have contacted the authors of [1] and will try to incorporate this in our framework to further extend our empirical evaluation.
>
> [1] Relative Flatness and Generalization, NeurIPS 202.
>
> >**Q1: How much does $w_q$ differ from $w$ in training?**
>
> Initially, the difference is relatively large when the weights gets clipped to quantization levels, but it decreases as training progresses and then remains almost constant throughout the rest of the training. Overall, the difference is relatively small and stays almost constant during training.
>
> >**Q3: Given a fixed quantization bit $b$, how would $s$ influence the performance?**
>
> We are not sure to understand the reviewer's question. To clarify, $b$ is pre-determined by the user and the channel-wise $s$ values are learned during training; as such, they can automatically adapt to the given $b$ value.
>
>
> >**Q4: What is the scale of w (e.g. norm) across different algorithms in section 3.3?**
>
> Following the reviewer's suggestion, we measured the norm of the different $w$s in the network and observed no significant difference between the norms of quantized and non-quantized parameters, even across different datasets.

---

> > ### Comment · Reviewer_SqJY · 2024-11-26
> >
> > Thank you for the response, which has clarified most of my doubts. Having a paragraph introducing how a model is trained for quantization would be helpful.
> >
> > I would like to maintain my original score.

---

> > > ### Author Response · Authors · 2024-11-26
> > > **Thank you!**
> > >
> > > We sincerely appreciate your thoughtful feedback and acknowledgment of the clarifications provided. We value your suggestion to include a paragraph on how a model is trained for quantization and will incorporate this, along with additional details, into the implementation section of the manuscript.
> > > If you have any further questions, concerns, or areas where additional clarification could assist in your evaluation and potentially lead to a higher rating, we would be more than happy to provide detailed responses.

---

### Official Review · Reviewer_NxQN · 2024-11-04

**Soundness:** 3
**Presentation:** 2
**Contribution:** 2
**Rating:** 5
**Confidence:** 4

**Summary:**

This paper investigates the impact of quantization noise on out-of-distribution (OOD) generalization. The authors observe that quantization noise appears to enhance OOD generalization capabilities in domain generalization tasks. While the observation is interesting, the methods employed for quantization and ensemble are standard and lack novelty.

**Strengths:**

This paper is well-organized in general, and the observation that quantization noise can improve OOD generalization is interesting.

**Weaknesses:**

1. The methods for quantization and ensemble are quite standard and not specific to domain generalization, providing limited novelty.
2. The theoretical justification linking flat minima to improved OOD generalization is weakened by recent research suggesting that the connection between flatness and generalization is questionable [1,2].
3. Quantization noise is similar to uniform weight noise, but a systematic comparison with the latter as well as other weight perturbation schemes is missing.

[1] Andriushchenko, Maksym, et al. "A modern look at the relationship between sharpness and generalization." arXiv preprint arXiv:2302.07011 (2023).
[2] Mueller, Maximilian, et al. "Normalization layers are all that sharpness-aware minimization needs." Advances in Neural Information Processing Systems 36 (2024).

**Questions:**

It would be interesting to see how different weight perturbation schemes can affect OOD generalization.

---

> ### Author Response · Authors · 2024-11-21
>
> We sincerely appreciate your recognition of the intriguing role of quantization noise in enhancing generalization, as well as your positive remarks on the organization of our work. Below, we address your major concerns in detail:
>
> >**W1: The methods for quantization and ensemble are quite standard and not specific to domain generalization, limited novelty**
>
> Our approach is the first to use quantization to address domain generalization challenges. We present novel analysis to show that quantization-aware training introduces an implicit regularization effect, guiding the model toward flatter minima and thus enhancing out-of-distribution (OOD) generalization. We appreciate that the reviewer acknowledges this as an ``interesting observation", which further highlights that it brings novelty to the community.
>
> Additionally, we introduce the Ensemble of Quantization (EoQ), which combines quantization and ensembling to amplify the benefits of both techniques. EoQ achieves state-of-the-art OOD performance while requiring minimal resource overhead, demonstrating its practical and methodological significance. Our work not only highlights a novel application of quantization noise but also provides empirical validation of its effectiveness across diverse datasets, setting it apart from standard techniques. We believe this to be of interest to the community.
>
> >**W2:How does your work reconcile the flatness-generalization relationship with recent critiques[1,2]?**
>
> Thank you for highlighting this important discussion!
>
> To better understand the conclusions of [1], we reached out to its authors, and Andriushchenko (the primary author) directed us to his post-ICML discussions (https://x.com/maksym_andr/status/1687395919442948096), where he acknowledges that sharpness (or its inverse, flatness) can still be a valuable measure of generalization. He states that their work ``doesn't imply that sharpness is useless, particularly since the empirical success of SAM is undeniable." This perspective aligns well with our findings. We do not claim that flatter minima are universally better, but instead demonstrate their utility in reducing sensitivity to domain shifts, as supported by experiments across multiple benchmarks.
>
> Furthermore, the work in [2] is orthogonal to our findings. It shows that, on certain datasets, applying SAM to normalization layers only can improve performance over applying it across the entire model. We added discussion around [1,2] in the revised manuscript to address the nuanced relationship between flatness and generalization.
>
> [1] Andriushchenko, Maksym, et al. "A modern look at the relationship between sharpness and generalization." ICML, 2023. \
> [2] Mueller, Maximilian, et al. "Normalization layers are all that sharpness-aware minimization needs." NeurIPS, 2023.
>
> >**W3: How does quantization noise compare to uniform weight noise and other weight perturbation schemes?**
> >
> Quantization noise and uniform weight noise share similarities in that both introduce perturbations to the model's parameters. However, quantization noise specifically arises from the discretization of the weights, which can lead to a more structured form of regularization due to the rounding or truncation during the quantization process. In contrast, uniform weight noise typically adds random perturbations with a uniform distribution, which may not exhibit the same structured regularization properties.
>
> Below, we provide the results of our ablation study on the PACS dataset with uniform noise with different minimum and maximum value:
>
> | Noise                        | OOD Accuracy      |
> |------------------------------|-------------------|
> | no noise                     | 84.7 ± 0.5        |
> | uniform(-0.0001, 0.0001)     | 82.8 ± 0.6        |
> | uniform(-0.00005, 0.00005)   | 83.9 ± 0.4        |
> | uniform(-0.00001, 0.00001)   | 84.9 ± 0.3        |
> | uniform(-0.000005, 0.000005)   | 85.4 ± 0.4        |

---

> ### Author Response · Authors · 2024-11-26
>
> We hope that our response have provided clarity and effectively addressed your concerns. We would greatly appreciate it if you could acknowledge this. If there are any remaining questions or unresolved concerns, we would be more than happy to provide further clarification. Thank you sincerely for your time and valuable feedback—it is greatly appreciated.

---

> > ### Author Response · Authors · 2024-12-02
> > **Thank you!**
> >
> > Dear reviewer,
> > As the discussion period nears its end, we kindly hope to receive your feedback today. We have carefully highlighted all revisions in red and would greatly appreciate your kind consideration before finalizing your recommendation.
> >
> > We greatly value your input and hope the updates meet your expectations.

---

### Official Review · Reviewer_Wsn5 · 2024-11-04

**Soundness:** 3
**Presentation:** 3
**Contribution:** 3
**Rating:** 6
**Confidence:** 4

**Summary:**

The paper works on domain generalization problem through weight quantization, which can be an implicit regularize by inducing noise in model weights to guide the optimization process to flatter minima against domain shifts. The paper provides both theoretical and empirical evidence for quantization to encourage flatter minima. Experiments on several datasets demonstrate the generalization ability of the method across various datasets, architectures, and quantization algorithms.

**Strengths:**

1. The paper is well-written and easy to follow.

2. The idea that improves the domain generalization ability by weight quantization is interesting.

3. The experimental results demonstrate the effectiveness of the proposed method.

**Weaknesses:**

1. Equation 4 shows that the weights in flatten minima should have lower losses on the quantization model. However, it is not clear how the loss function guarantees that lower loss can in turn lead to local minima. It is also not clear how the large noise dominate the optimization and lead to sub-optimal convergence.

2. As shown in the experiments, the model size will definitely be reduced through quantization, but how about the training and inference costs of the model? Especially when the quantization is conducted during the model training.

**Questions:**

1. How are the ensemble quantization models trained? The models are trained differently from the beginning or from the quantization step?

2. How does the quantization step (e.g., 2000 for DomainNet) affect the model performance? How do the author select such hyperparameter?

---

> ### Author Response · Authors · 2024-11-21
>
> Thank you for appreciating the clarity of our writing, the novelty of leveraging weight quantization for domain generalization, and the strength of our experimental results. We address your major questions and concerns below.
>
> >**W1: Equation 4.**
>
> Thank you for your comment. We acknowledge that our discussion of Equation 4 might have been unclear. In short, and as updated in the paper, Equation 4 shows that the noise induced by the quantization process will facilitate the optimization to escape from sharp minima and instead settle in flatter ones. There is however no guarantees that these flatter minima have a lower loss value than others or are global minima. We hope the revised text for Equation 4 clarifies what we meant.
>
> >**W2: How does large noise dominate optimization in Equation 4 and result in sub-optimal convergence?**
>
> The quantization-induced noise $(\Delta\)$ in Equation 4 affects the loss function as follows:
>
> $L(w + \Delta) \approx L(w) + \nabla L(w)^T \Delta + \frac{1}{2} \Delta^T \mathcal{H} \Delta,$
>
> where $\mathcal{H}$ is the Hessian matrix. The second and third terms in this approximation introduce a regularization effect: they penalize sharp minima (large eigenvalues of $\mathcal{H}$) and encourage convergence to flatter regions of the loss surface. However, if the noise $(\Delta\)$ becomes too large, it introduces over-regularization. This excessive noise can overly restrict the search space, preventing the model from reaching a good solution. Instead, the optimization process may focus on minimizing the loss in a way that avoids sharp regions, but sacrifices the ability to find true minimum of the loss function.
>
> Table 8 evidences this trade-off: Moderate noise improves out-of-distribution (OOD) generalization, but excessive noise harms both in-domain and OOD performance. The noise perturbation introduces stochasticity into the optimization process, which biases the trajectory toward flatter minima, but too much noise destabilizes the process, leading to suboptimal convergence.
>
> We have revised the text in the paper to clarify this.
>
>
> >**W3: How does quantization affect model size, training costs, and inference costs?**
>
> Quantization reduces both the memory footprint and latency. For example, a ResNet-50 model running on an AMD EPYC 7302 processor achieves a latency of 34.28ms in full precision and 21.02ms with 8-bit quantization. While quantization can theoretically reduce training costs by enabling dynamic bit precision switching, our setup does not support this, so training time remains almost unchanged. However, inference time and model size are significantly reduced, as detailed in the results section of the paper.
>
> >**Q1: How are ensemble quantization models trained—differently from the beginning or only at the quantization step?**
>
> The models are trained independently from initialization, using different random seeds to ensure diversity. We have clarified it in the updated manuscript.
>
> >**Q2: How does the quantization step affect the model performance? How do the author select such hyperparameter?**
>
> The quantization steps were empirically determined. We added the ablation studies in the appendix to discuss the performance trends. Here is the ablation study performed on the PACS dataset:
>
> | Quantization Step | OOD Accuracy      |
> |-------------------|-------------------|
> | No quantization   | 84.7 ± 0.5        |
> | 1000              | 86.2 ± 0.4        |
> | 2000              | 87.8 ± 0.3        |
> | 3000              | 86.9 ± 0.4        |
> | 4000              | 85.1 ± 0.3        |

---

> ### Comment · Reviewer_Wsn5 · 2024-11-25
>
> Thanks for your responses. Most of my concerns have been addressed and I would like to maintain my original rating of 6.

---

> > ### Author Response · Authors · 2024-11-25
> > **Thank you!**
> >
> > We sincerely appreciate your acknowledgment and recommendations. If there are any additional questions or concerns that we can address to further assist you in evaluating our work and potentially raising your rating, we would be happy to provide detailed responses.

---

### Meta-Review · Area_Chair_KkSt · 2024-12-19

**Metareview:**

The submission proposes a new approach for addressing the domain generalisation problem, where one must train a model on data from several source domains with the goal of zero-shot generalisation to one or many target domains. The proposed approach leverages weight quantisation, and the success of this method is explained by appealing to flat minima. The submission also includes an exploration of ensemble approaches to the DG problem, showing that this improves performance.

The reviewers have expressed concern about the novelty of the proposed approach, as it is a relatively straightforward combination of existing quantisation and ensemble approaches. Moreover, the connection between flat minima and OOD generalisation performance is not made clear.

**Additional Comments On Reviewer Discussion:**

There was some discussion between the authors and reviewers, but this did not result in much change in opinion.

---

### Decision · Program_Chairs · 2025-01-22

Reject